# Creating frames of reference for chert exploitation during the Late Pleistocene in Southwesternmost Iberia

**Joana Belmiro**[1]*, **Xavier Terradas**[2], **João Cascalheira**[1]

**1** Interdisciplinary Center for Archaeology and the Evolution of Human Behavior (ICArEHB), University of Algarve, Faro, Portugal, **2** Consejo Superior de Investigaciones Científicas (CSIC), Madrid, Spain

* jfbelmiro@ualg.pt

## Abstract

Southwestern Iberia has played a key role in characterizing Late Pleistocene human ecodynamics. Among other aspects of human behavior, chert procurement and management studies in this region have received increasing attention in the past two decades, especially focusing on the sites showing repeated human occupation, such as the case of Vale Boi (Southern Portugal). However, these studies have been very limited in their geographical scope, and mostly focused on brief macroscopic descriptions of the raw materials. To further our knowledge of the relationship between regional availability of raw materials and its impact on human adaptations and mobility, a more detailed approach to characterizing geological sources is needed. This paper characterizes chert raw materials location, diversity, and availability in a geologically well-defined region of southern Portugal - the Algarve. Through macroscopic and petrographic approaches, we provide a detailed characterization of geological chert sources to build a frame of reference for chert exploitation in the region. Our results show that there are four main chert formations in Algarve, and that despite the within-source variability, sufficient differences at macroscopic and petrographic levels are present to allow clear source attribution. These results provide a baseline for raw material studies in archaeological assemblages across southwestern Iberia, that will be essential to further characterize the dynamics of human behavior in some of the most important eco-cultural niches.

## Introduction

Southern and Western Iberia have often been considered key areas to understanding techno-cultural transitions from the Middle Paleolithic to the end of the Upper Paleolithic. As territories located at the western tip of the European continent and with a generally stable climate even during the coldest periods punctuating the Late Pleistocene, these have been regarded as some of the most significant glacial refugia in Europe [1–4]. For this reason, Southwestern Iberia has frequently been at the center of some of the most debated topics regarding Late Pleistocene human adaptations [5–7]. A particularly good example is the region's role in possibly

**Data Availability Statement:** The complete R code used for all the analysis and visualizations contained in this paper, as well as supporting

materials, are available at our online research compendium (DOI: 10.17605/OSF.IO/FP7TA).

**Funding:** J.B. is funded by Fundação para a Ciência e Tecnologia (https://www.fct.pt/), contract reference 2020.08722.BD. J.C. is funded by Fundação para a Ciência e Tecnologia (https://www.fct.pt/), contract reference DL57/2016/CP1361/CT0026. Current work on chert identification in Algarve is being supported by the European Research Council through the FINISTERRA project (101045506-FINISTERRA-ERC-2021-COG) attributed to J.C. The funders had no role in study design, data collection and analysis, decision to publish, or preparation of the manuscript.

**Competing interests:** The authors have declared that no competing interests exist.

being one of the last territories to be occupied by Neanderthal populations right before their complete disappearance [6, 7]. Neanderthals occupied the European continent for more than 300.000 years and are thought to have disappeared while modern humans arrived on the territory [6, 8–11]. The last territory where Neanderthal populations seemed to exist was Southern Iberia, around c. 37 thousand years ago [7], or perhaps even earlier [1, 6, 12]. This region is key to understanding how Neanderthals survived until such a later chronology, the degree and types of interaction those populations may have had with modern humans [6, 7], and how and why they eventually went extinct [13, 14]. Another example is Southwestern Iberia's importance in the discussion of Upper Paleolithic technocomplexes transitions. Previous studies have discussed the territory's potential as a refugium during cold and harsh climatic conditions [2, 8, 15]. The Heinrich Event 2 (HE 2) at the onset of the Last Glacial Maximum (LGM), for example, is a period marked by important social and technological transformations. This climatic event was characterized by abrupt and drastic climatic changes that impacted human behavior all across westernmost Europe [16, 17]. The identification of a Proto-Solutrean phase in central and southern Portugal with a very distinct index fossil (the Vale Comprido point), and its direct association with the HE 2 [18], put these regions amongst some of the most important case studies of how environmental dynamics have affected human adaptations during the last glacial. Other studies have expanded upon this notion of climatic refugia during harsh climatic events, to understand the Iberian Peninsula as a long-term eco-cultural refugia [5]. Using this framework, this territory would consist of several ecological niches, consistently used through time, possibly due to the stability in the richness and variety of resources. This continuous use would then create long-term regional adaptive structures, which when correlated with the ecological niches, Cascalheira et al. [5] have referred to as eco-cultural niches. In fact, it seems that a large number of caves and rockshelters in Iberia are multi-layered, giving validity to the aforementioned framework [19]. These eco-cultural niches provide an exceptional opportunity to understand long-term dynamics regarding biotic and abiotic resource exploitation since they can provide details on how human populations maintained or changed their adaptive systems when facing environmental changes, and cultural and social transformations or constraints. One of such possible Late Pleistocene eco-cultural niches to which this theoretical framework has been previously applied is the archaeological site of Vale Boi. This multi-component site is located in westernmost southern Iberia, in a region currently known as the Algarve, and comprises one of the most complete Upper Paleolithic chronocultural sequences of southern Iberia [5, 20, 21]. Several cross-scale complex interactions have been identified, displaying resilient behaviors throughout the Upper Paleolithic maintained by their eco-cultural niche, but also adaptation behaviors motivated by niche diversity, social networks, and climatic changes [5]. Some of these resilient elements are, for example, the continuous use of strategies like grease rendering and selective hunting patterns, site function, certain lithic technology patterns, and the functional specialization of lithic raw materials. The maintenance of social networks through the identification of possible long-distance lithic raw materials has also been suggested [22]. The identification of these patterns has been reliant on the great amount of studies that have originated from the archaeological site of Vale Boi [i.e., 22, 23, 24, 25]. A large portion of these studies has focused on lithic technology [26–28], and unlike other regions of Portugal [29–35], archaeological studies focusing on raw materials, and especially chert, have been more scarce, focusing mostly on brief macroscopic results or the differentiation between possibly local and non-local sorts of raw materials. Despite the scarceness of these studies, chert played a significant role in early and later prehistory at Vale Boi [26, 27, 36], and across most of Prehistory in Southern Portugal [37, 38]. As an essential part of late Pleistocene hunter-gatherer adaptations, lithic raw materials have the potential to provide insights into the adaptive strategies of those populations [39–41], mostly regarding land-use,

technological organization, but also cultural and social interactions. In fact, the selection and procurement of raw material has been suggested as a key stage for the technological organization of hunter gatherer groups [42]. Thus, changes in the frequencies of raw materials within the archaeological records may provide evidence for changes in that organization, or even resilience of specific choices, all of which can reflect culturally transmitted preferences within a group of hunter gatherers [43]. Several models, both formal and informal, have shown that changes of raw material in the archaeological record may reflect different procurement strategies [44–47], either related to mobility strategies [48–51] or changes in the availability of raw materials, possibly related to environmental change [43, 52]. Raw materials are also closely linked to the tools produced with them–the costs of manufacturing technology can be related to the availability of raw materials in the landscape [53–56], or the characteristics of the raw materials which might relate to functionality [57–59], through the choice of raw materials for specific technologies [43]. Furthermore, changes in the raw materials can also be related to the establishment of social networks [60] and the horizontal transmission of preferences through trade [43]. In order to explore the aforementioned theories and models in an archaeological site, especially when trying to understand land-use and the adaptive strategies of an eco-cultural niche like what is proposed for southwesternmost Iberia, it is necessary to know the landscape and the raw material sources available in the territory. To distinguish between local and non-local raw materials, first, it is necessary to understand the characteristics of the local sources and establish a comparative database that can be used for the analysis of an assemblage. This is true for most of the informal models which try to understand why raw materials change in the archaeological record. Formal models focusing on raw material use posit that it is necessary to have detailed sourcing data in order to adequately apply the models to a region or an archaeological assemblage [61, 62]. In fact, the creation of such a database is a starting point for most raw material-focused studies, independent of the geography or studied chronology [63–66]. In western Europe, there have been substantial efforts in creating comprehensive knowledge about chert-bearing formations, which resulted in important lithotheques and databases [67–69], and multidisciplinary studies of raw material use throughout prehistory [29, 70–74], as early as the 1980s.

In southern Portugal, previous efforts have been made to create such a database. The work of Verissimo [75], albeit focusing on the occurrence of chert solely in western Algarve, provided the initial basis for comparative studies with the assemblages from some Late Pleistocene sites based on macroscopic approaches. The creation of LusoLit, a lithotheque currently hosted at the University of Algarve [76], and the collection of samples from several outcrops in the region provided a new leap in the study of chert in the region. A few geological studies have also contributed to understand the availability and characteristics of chert in southern Portugal [77, 78]. Nevertheless, these studies are often unpublished or comprehend answers to geological questions, which hamper the comparative use of the data with archaeological assemblages. A large portion of chert-bearing outcrops in southern Portugal remains unstudied, both by archaeologists and geologists. As such, further analyses of the overall variability, location, and availability of chert nodules in the Algarve are necessary to start testing behavioral models of land-use and abiotic resource exploitation. Given the potential for an in-depth raw material study at a possible Late Pleistocene eco-cultural niche such as Vale Boi—also due to the ubiquitous presence of chert throughout the stratigraphy, as one of the main raw materials used for lithic technology in the site [26, 27, 36]—in this study we explore the location, diversity, and availability of chert raw materials in the southernmost region of Portugal–the Algarve. Our main goal is to establish a reference for chert raw materials in an understudied region in regards to lithic materials, that can be used for future studies addressing chert exploitation in the Algarve and elsewhere. This includes the development of a methodological approach

adapted to the study area, by testing the potential of macroscopic and petrographic approaches for the characterization of regional cherts. The data presented here is also fully integrated into an online lithotheque (LusoLit) that is now freely available and can be built upon and improved as new data becomes available.

## Geological setting and chert outcrops

### Geological setting

The Algarve is the southernmost region of Portugal, framed north by the Alentejo region and east by Spain. To the west and south, it is bordered by the Atlantic ocean. It extends for ~130 km E-W and ~50 km N-S and is characterized by a variety of geomorphic sub-regions and geological units, that make this region a complex territory (Fig 1). On the north sector of the Algarve, the Serra Algarvia is characterized by a mountainous range with a dense hydrographic network, which separates the Algarve from Alentejo. On the south sector, the Litoral is characterized by a flatter, long strip of land, that extends through all of the coastal strip of the Algarve. The Barrocal is nested between the other sub-regions and has a more moderate relief, characterized by carbonated Jurassic formations and important sub-terranean water circulation [79].

Geologically, the Algarve is composed of two main geological units: the South Portuguese Zone (SPZ) and the Algarve basin. The SPZ is located in the north sector of the Algarve, extending up to Alentejo [80]. Its main lithologies are schist, greywacke and quartzite [22, 80]. The SPZ is overlain unconformably by the Mesozoic sedimentary rocks of the Algarve basin [80]. The basin corresponds to the Mesozoic-Cenozoic sediments that outcrop south of the Algarve, from the westernmost to the easternmost point of the region, and it is associated with the opening of the central Atlantic Ocean and with the eventual oceanic crust formation in the western part of the Tethys sea, between the Algarve and North Africa [81]. Mesozoic sedimentation of the basin started in the Triassic and continued thereon. In the Lower Jurassic (Lower Pliensbachian, also regionally known as Carixian) the basin was divided into two sub-basins–western and eastern sub-basins [78, 81]. The existence of the two sub-basins and the expansion and retraction of the sea created a variety of sedimentation environments, such as external and internal platforms, continental, hemipelagic and deep marine [81], as well as moments of sedimentation hiatus. This variability in deposition environments created a variety of sedimentary facies, with moments of more or less homogeneity throughout this period. For example, during the Lower Pliensbachian, in the Lower Jurassic, the sediments in the western sub-basin can be described as marine of external platform, while the sediments of the eastern sub-basin can be described as marine of internal platform. During the Upper Jurassic however, the basin is marked by a moment of prominent lithofacies variation, followed by a moment of uniformity in both sub-basins [81]. Understanding the Algarve basin is key for raw material studies in the Algarve, especially when studying chert, since it is in the basin, more specifically in the Jurassic sediments, where chert primarily outcrops in the region.

### Chert outcrops

Within the Algarve area, the presence of chert may be associated with carbonates in limestone and dolomite formations. This is explained by characteristics (such as the presence of water or specific temperatures and pH) that are ideal for both the formation of limestone and the precipitation of silica [66]. The pelagic and marine environments of the Algarve basin during the Jurassic gathered those such ideal characteristics: as shown by Ribeiro [77], the cherts from the Lower Jurassic are the result of early diagenic silicification of carbonate sediments. The existence of two basins with different sedimentation environments also shows potential for the existence of different types of chert throughout the basin and their differentiation. For

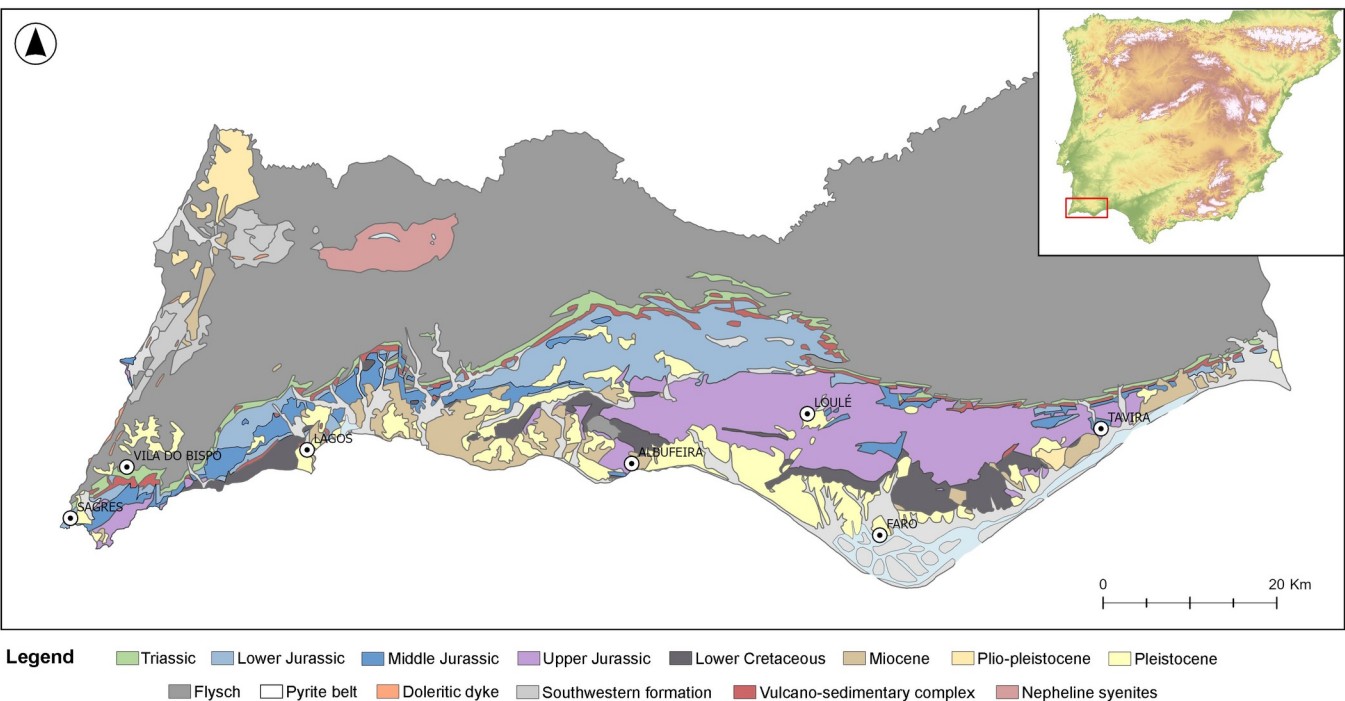

**Fig 1. Geological map of the Algarve region.** The map represents the several geological layers and formations, as well geomorphic sub-regions. The geological data was obtained through the vectorization of the geological raster maps (Carta Geológica de Portugal, 1:500 000 scale) made available by LNEG (Laboratório Nacional de Energia e Geologia) in https://geoportal.lneg.pt/pt/dados_abertos/cartografia_geologica/cgp500k/folhasul/. The inset map was created from a 30×30-m pixel Digital Elevation Model (DEM) obtained from the Shuttle Radar Topography Mission (https://www2.jpl.nasa.gov/srtm/dataprod.htm).

example, during sedimentation, skeletal grains of fossils may be preserved. Many of these fossils are restricted to specific environments and time intervals [82], which may allow the identification of chert outcrops through the fossil content. The basin and sub-basins thus show potential for the existence of different geological formations with different chert types and their study. Previous works, both geological and archaeological, confirm that chert is present in the Algarve in the Jurassic limestone or dolomitic limestone layers of the Algarve basin. This means that chert outcrops can be identified in the central/south sector of the region, from west to east. Variability in chert availability, as well as chert types, is expected, considering that during the sedimentation process, the Algarve basin was already sub-divided and in constant environmental change. Due to this, several formations with chert nodules can be identified in the Algarve, attributed to different sub-periods of the Jurassic. In the western sector of the Algarve, chert can be found in the Lower Jurassic (Carixian) formations, in limestone or dolomitic limestone layers [78, 83], often visible in areas where the layers are exposed, such as beach-generated cliffs and associated deposits, like Cabo de S. Vicente (CSV) and Praia do Belixe (PBX). These formations with chert nodules are also visible inland, albeit more scarcely, as is the case of the small outcrop named Ferrel (FER), 3 km from the current coastline. Lower Jurassic chert-bearing formations are barely existent in the center/east sector of the Algarve, with one single formation with micro-nodules identified in geological works [84]. Middle Jurassic geological layers with chert nodules are only found in the center/east sector of the Algarve in a geological formation called the Malhão formation. The formation can be described as carbonated, from a marine sedimentation environment. Chert in this formation has been identified in two distinct layers: conglomerates with micritic limestone intercalations with chert beds and nodules, characterized by the presence of sponge spicules and radiolarians

[85]; microcrystalline limestones with chert nodules characterized by the presence of silicified malacofauna and silicified corals [86]. Finally, Upper Jurassic sediments with chert nodules have also been mostly identified in the center/east sector of the Algarve, attributed to the Jordana formation. This formation is characterized by dark-gray limestones, with frequent secondary silicifications with abundant fossil fragments [85–87]. Upper Jurassic sediments with chert nodules in western Algarve (Kimmeridgian formations) have only been identified in one area, between Ponta da Atalia (PtA) and Praia da Mareta (MAR) [83]. Given the differences of the cherts and formations between the western and eastern sectors of the Algarve already established in previous works, this division will be followed in the present study.

## Materials and methods

To locate and characterize chert formations and corresponding outcrops in southern Portugal and understand the chert's characteristics, a macroscopic and petrographic approach was applied to the study of geological samples which were collected through fieldwork. Combining different analyses and methods provides a comprehensive approach to reconstructing the geological and geographical origin of raw materials, especially since different methods have their inherent limitations [66]. Several other similar methodologies and approaches have been applied in other regions [70, 88–90]. However, the chosen analysis techniques should be adapted to the specific geographic context, the research questions, the problematics, and the characteristics of the types of cherts in question [66]. Since only preliminary studies of raw materials were applied in the western portion of southern Portugal, and petrographic data has been shown to provide good results for the characterization of cherts in this region [77], the two methodologies were chosen for the study. The geological samples used in this study were obtained during fieldwork, between August 2021 and June 2022. The prospected locations were chosen after reviewing previously known research, which included preliminary raw materials studies in the region [22, 75], geological scientific papers and theses focusing on the Algarve basin and concerning chert-bearing formations and lithologies [77, 78, 91, 92], and geological maps, which signaled the presence of chert nodules within the formations and geological layers [83–87, 93, 94]. Unpublished data and coordinates for unsurveyed locations with potential for chert-bearing outcrops gathered during the organization of the LusoLit lithotheque were also checked. Whenever coordinates or specific locations for known outcrops were available, these were directly visited and the surrounding area was surveyed to understand the extension of the outcrops and to locate possible secondary deposition outcrops nearby. When no specific locations within a formation were described (for example, in geological maps) several areas with more potential to find chert outcrops within one formation were surveyed. Samples were collected whenever possible, focusing on both primary and secondary outcrops. When chert nodules within one single outcrop showed macroscopic differences (such as differences in the color, texture, translucency, or cortex), samples of each different nodule were collected, to cover all chert variability within the outcrop, and understand chert variability within the formation. This variability was also recorded using a database (to distinguish between homogeneous or heterogeneous chert nodules within the outcrop) and through photography (Fig 2). All samples were registered with resource to a free android app (Archaeosurvey) which was designed for archaeological surveys, and records site location and characteristics [95], further adapted for raw material source surveys [96]. The version of the software used for fieldwork is an adaption of the latter apps and records data related to outcrop characteristics and conditions (i.e. abundance, visibility, access, geomorphology, chert morphology, and conditions). All data related to the app and fieldwork can be found in the Supplementary Online Materials (S1 Table). Individual IDs

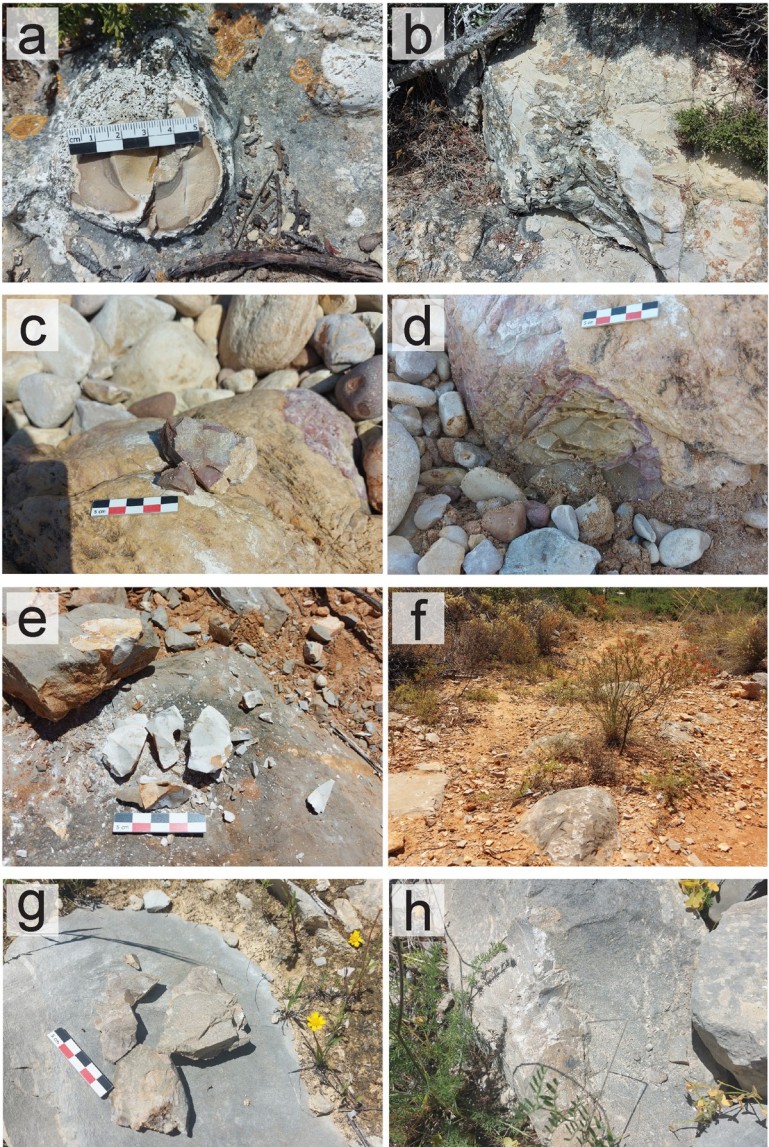

**Fig 2. Recovered geological chert samples and general outcrop photos from chert-bearing formations, collected during fieldwork (2021–2022).** (a) Detail of a chert nodule (SP32_FZF). (b) Chert outcrop Foz dos Fornos (FZF) (Lower Jurassic, Carixian formation) associated with sample SP32_FZF. (c) Detail of a chert nodule (SP69_MAR). (d) Chert outcrop of Praia da Mareta (MAR) (Upper Jurassic, Kimmeridgian formation) associated with sample SP69_MAR. (e) Detail of recovered chert samples (SP65_MALH) and nodules. (f) General photo of the Malhão outcrop (MALH) (Middle Jurassic, Malhão formation) associated with sample SP65_MALH. (g) Detail of recovered chert samples (SP58_JOR). (h) General photo of the Jordana outcrop (JOR) (Upper Jurassic, Jordana formation) associated with sample SP58_JOR.

were associated with each sample, which includes sequential numbers (based on recovery order, i.e., SP10) and outcrop code (i.e., PdA).

A two-step approach was applied to characterize the geological samples. These include 59 geological samples (S2 Table), currently located at ICArEHB's laboratories (University of Algarve). No permits were required for the described study, which complied with all relevant regulations. The samples were analyzed macroscopically following a pre-established dataset. The variables were defined based on specialized literature [66, 97, 98], and the dataset with the

sample characterizations as well as the variable descriptions can be found in the SOM (S2 and S3 Tables). A small hand lens of 10x magnification was used for this analysis, followed by a higher magnification analysis with resource to a Nikon SMZ25 stereomicroscope, focusing primarily on inclusions and fossil content. Despite several caveats, especially related to the subjectivity and lack of quantitative variables [63], a macroscopic approach is currently still frequently used in chert raw material studies. For a comparative analysis of archaeological artifacts, other methods may be inconvenient or impossible to use, since they may be destructive and often difficult to apply to large assemblages. Macroscopic analyses have the advantage of being less costly and easy to apply. Establishing a reliable macroscopic characterization and understanding the potential of macroscopy to differentiate between cherts, outcrops and formations is essential for comparative studies between geological samples and archaeological assemblages and obtaining preliminary results. By then combining the macroscopic analysis with a petrographic analysis, other studies have shown that the subjective component of this approach can be minimized [63]. The second phase of the study focused on the petrographic analysis of the geological samples. Thin sections were produced from geological samples of all formations, focusing on obtaining petrographic data that reflected the macroscopic variability. In total, 30 thin sections were produced (Table 1), divided into three groups: 1) 20 thin sections of geological samples from different outcrops of the Lower Jurassic and Upper Jurassic chert-bearing formations within the western section of the Algarve; 2) 9 thin sections of geological samples from different outcrops of the Malhão and Jordana chert-bearing formations, from the eastern section of the Algarve; 3) 1 thin section of a geological sample recovered from previous works, which was not identified during our survey. Although primary outcrops were prioritized, thin sections of secondary deposition samples were also produced. To compare with the thin sections from this study, other thin sections from previous studies of Jurassic outcrops from western Algarve [77] were also consulted. All thin sections were analyzed using a Nikon LV100ND or a Leica DM2500 P and following standard petrographic description (full descriptions of the variables considered for the petrographic description can be found in S3 Table).

All sample descriptions (macroscopic and petrographic) and accompanying photographs, as well as photographs and data about the outcrops are also available on a database dedicated to the LusoLit lithotheque, which can be accessed online at https://lusolit.icarehb.com/. This database will continue to be updated with the other existing samples, non-chert raw materials and future analyses. The complete R code used for all the analysis and visualizations contained in this paper is available at our online research compendium (https://doi.org/10.17605/OSF.IO/FP7TA). To produce those files, we followed the procedures described by Marwick [99] for the creation of research compendiums to enhance the reproducibility of research. The files provided contain all the raw data used in our analysis as well as a custom R project [100] holding the code to produce all tables and figures. To enable maximum reuse, code is released under the MIT license, data as CC-0, and figures as CC-BY (for more information, see [99]).

## Results

Eighteen outcrops (primary and secondary) were revisited or newly identified in the Algarve region. Nine are located in the westernmost territory and nine to the east (Fig 3). From these, 57 samples were recovered and analyzed, of which 19 are isolated finds or in secondary settings (Table 2).

### Western Algarve

On the westernmost areas of the Algarve, there are mainly cherts from two different formations: Carixian formations (Lower Jurassic) and Kimmeridgian formations (Upper Jurassic,

**Table 1. List of geological samples chosen for the petrographical study.**

| Sample ID | Type | Laboratory | Outcrop | Formation | Epoch | Sample collection |
|---|---|---|---|---|---|---|
| SP6 | Covered | UB | Cabo S. Vicente | Carixian | Lower Jurassic | 2021 |
| SP7 | Covered | UB | Cabo S. Vicente | Carixian | Lower Jurassic | 2021 |
| SP9 | Covered | UB | Ponta dos Altos | Carixian | Lower Jurassic | 2021 |
| SP10 | Covered | UB | Ponta dos Altos | Carixian | Lower Jurassic | 2021 |
| SP14 | Covered | UB | Praia Belixe | Carixian | Lower Jurassic | 2021 |
| SP15 | Covered | UB | Praia Belixe | Carixian | Lower Jurassic | 2021 |
| SP18 | Covered | UB | Praia Belixe | Carixian | Lower Jurassic | 2021 |
| SP21 | Covered | UB | Belixe Sul | Carixian | Lower Jurassic | 2021 |
| SP23 | Covered | UB | Belixe Norte | Carixian | Lower Jurassic | 2021 |
| SP24 | Covered | UB | Cabo S. Vicente | Carixian | Lower Jurassic | 2021 |
| SP27 | Covered | UB | Cabo S. Vicente | Carixian | Lower Jurassic | 2021 |
| SP28 | Covered | UB | Aspa | Carixian | Lower Jurassic | 2021 |
| SP32 | Covered | UB | Foz dos Fornos | Carixian | Lower Jurassic | 2021 |
| SP33 | Covered | UB | Foz dos Fornos | Carixian | Lower Jurassic | 2021 |
| SP34 | Covered | UB | Ponta dos Altos | Carixian | Lower Jurassic | 2021 |
| SP34 | Covered | UB | Ponta dos Altos | Carixian | Lower Jurassic | 2021 |
| SP36 | Covered | UB | Ponta da Atalaia | Kimmeridgian | Upper Jurassic | 2021 |
| SP39 | Covered | UB | Andorinha | Kimmeridgian | Upper Jurassic | 2021 |
| SP40 | Covered | UB | Ferrel | Carixian | Lower Jurassic | 2021 |
| SP42 | Covered | UB | Boca do Rio | N/A | N/A | 2021 |
| SP59 | Covered | TSL | Jordana | Jordana | Upper Jurassic | 2022 |
| SP61 | Covered | TSL | Peral | N/A | Upper Jurassic | 2022 |
| SP56 | Covered | TSL | Jordana | Jordana | Upper Jurassic | 2022 |
| SP54 | Covered | TSL | Guilhim | Malhão | N/A | 2022 |
| SP53 | Covered | TSL | Guilhim | Malhão | Middle Jurassic | 2022 |
| SP58 | Covered | TSL | Jordana | Jordana | Upper Jurassic | 2022 |
| SP55 | Covered | TSL | Caliços | Malhão | Middle Jurassic | 2022 |
| SP50 | Covered | TSL | Casal da Colina | Malhão | Middle Jurassic | 2022 |
| SP52 | Covered | TSL | Casal da Colina | Malhão | Middle Jurassic | 2022 |
| RT82 | Covered | TSL | Praia Belixe | N/A | N/A | <2021 |

UB–Servei de Làmina Prima, University of Barcelona (Barcelona, Spain); TSL - Thin Section Lab (Toul, France).

Fig 3). The latter can be found in primary deposition in a single known outcrop - Praia da Mareta (MAR)–or nearby, in secondary deposition settings. Lower Jurassic outcrops are more common and, for that reason, have been more studied [77]. These outcrops are heterogeneous, showing different geological characteristics and chert colors (Fig 4). The Lower Jurassic cherts can be grouped into three main macroscopic types based on color (individual Munsell Color Chart codes can be found in the macroscopic description analysis table) and presence of fossil content: 1) multicolored, yellow, red, light gray or purple type (MC, Fig 4B and 4D); 2) single grey/brown type (SGB, Fig 4E and 4F); 3) multicolored, yellow, red, light gray or purple with fossils type (MCF, Fig 4A and 4C). The first two types are present in all outcrops. They are mainly characterized by dull to medium luster and opaque translucency, although some samples were sub-translucent. The feel ranges between smooth and semi-smooth, although many of the cherts from the Belixe outcrop are distinctly rough to the touch. In the MC cherts, fossil content is present but visible only as white, red, or yellow speckling. The SGB cherts show little fossil content, barely visible with the stereomicroscope. The MCF show a large quantity of

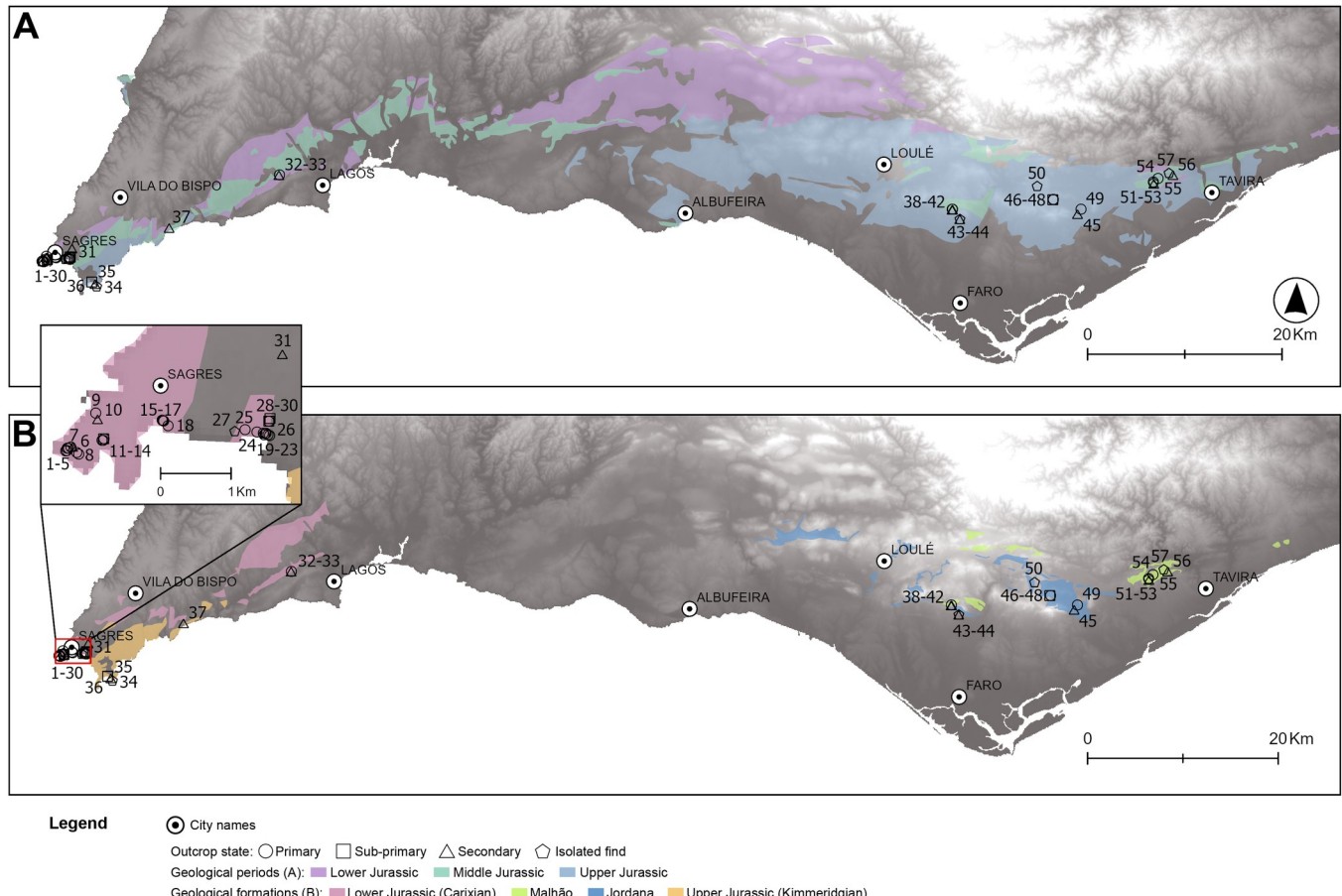

**Fig 3. Map of southern Portugal (Algarve), with geological samples recovered during this study's fieldwork.** Colors represent the chert-bearing formations in the Algarve. Numbers represent the recovered samples during fieldwork organized by formation, outcrop, and location. Lower Jurassic (Carixian) formation: 1–8: Cabo de S. Vicente(CSV); 9–10: Aspa (ASP); 11–14: Foz dos fornos (FZF); 15–18: Ponta dos Altos (PdA); 19–27: Praia de Belixe (PBX); 28–30: Belixe Sul (BLS); 32–33: Ferrel (FER). Upper Jurassic (Kimmeridgian) formation: 35: Ponta da Atalaia (PtA); 36: Praia da Mareta (MAR); 37: Praia da Andorinha (AND). Malhão formation: 38–42: Casal da colina (CdC); 43: Guilhim (GUI); 45: Caliços (CAL); 51–53: Oliveiras (OLV); 54–55: Malhão (MALH). Jordana formation: 46–49: Jordana (JOR); 50: Peral (PER). Isolated finds: 31: Belixe Norte (BLN); 34: Ponta da Atalaia (PtA); 44: Guilhim (GUI); 56: Descampado (DESCAM); 57: Pedreira (PEDR). The geological layers were obtained through vectorization of the geological raster maps (Carta Geológica de Portugal, 1:500 000 scale) made available by LNEG (Laboratório Nacional de Energia e Geologia) in https://geoportal.lneg.pt/mapa/. The base maps were created from a 30×30-m pixel Digital Elevation Model (DEM) obtained from the Shuttle Radar Topography Mission (https://www2.jpl.nasa.gov/srtm/dataprod.htm).

larger fossils (~1000 μm), which are easily seen by the naked eye and can be identified under the microscope.

Petrographically, the Lower Jurassic cherts of western Algarve are composed mainly of microcrystalline quartz, with textures that range mostly from wackestone to packstone (Fig 5). Dolomite is also present although in frequencies inferior to 10%. Macrocrystalline quartz and chalcedony occur in small frequencies (~5%), frequently replacing bioclasts. The presence of mica is uncommon and always below 1%. Allochems present are mostly iron oxides (ranging from uncommon to very frequent) and the presence of peloids is rare. In more than 50% of the samples, no fossils can be identified, as all fossils, albeit common to very frequent, are poorly preserved, and without any identifiable morphology. Whenever identifiable, fossils present in the sample are Echinoderms (Fig 5B and 5C), Radiolarians, Sponge spicules, and bivalve shells (Fig 5E and 5F). In all samples porosity ranges from 1–5% (vuggy type).

**Table 2. Outcrop description and chert information for all recovered samples.**

| Sample ID | Outcrop name | State | Age | Chert morphology | Chert frequence | Chert variability | Chert size |
|---|---|---|---|---|---|---|---|
| SP3_CSV | Cabo S. Vicente | Primary | Lower Jurassic | Nodule/Bedded | Sporadic | Homogeneous | 2-5cm |
| SP4_CSV | Cabo S. Vicente | Primary | Lower Jurassic | Bedded | Abundant | Homogeneous | - |
| SP6_CSV | Cabo S. Vicente | Primary | Lower Jurassic | Bedded | Abundant | Variable | - |
| SP7_CSV | Cabo S. Vicente | Primary | Lower Jurassic | Nodule/Bedded | Abundant | Homogeneous | - |
| SP8_PdA | Ponta dos Altos Este | Primary | Lower Jurassic | Nodule | Sporadic | Variable | 5-20cm |
| SP9_PdA | Ponta dos Altos Este | Primary | Lower Jurassic | Nodule | Abundant | Variable | 5-20cm |
| SP10_PdA | Ponta dos Altos Este | Primary | Lower Jurassic | Nodule | Abundant | Homogeneous | 5-10cm |
| SP12_PBX | Praia do Belixe | Primary | Lower Jurassic | Nodule | Sporadic | Homogeneous | - |
| SP13_PBX | Praia do Belixe | Primary | Lower Jurassic | Nodule | Abundant | Variable | Max. 15cm |
| SP14_PBX | Praia do Belixe | Primary | Lower Jurassic | Nodule/Bedded | Abundant | Variable | Max. 15cm |
| SP15_PBX | Praia do Belixe | Primary | Lower Jurassic | Nodule/Bedded | Abundant | Homogeneous | Max. 20cm |
| SP16_PBX | Praia do Belixe | Primary | Lower Jurassic | Nodule | Abundant | Homogeneous | 3cm |
| SP17_PBX | Praia do Belixe | Primary | Lower Jurassic | Nodule | Abundant | Variable | 5–15 cm |
| SP18_PBX | Praia do Belixe | Primary | Lower Jurassic | Nodule/Bedded | Abundant | Variable | - |
| SP19_PBX | Praia do Belixe | Primary | Lower Jurassic | Nodule | Rare | Variable | 5-20cm |
| SP20_BLS | Belixe Sul | Sub-primary | Lower Jurassic | Block | Abundant | Variable | - |
| SP21_BLS | Belixe Sul | Sub-primary | Lower Jurassic | Nodule | Abundant | Variable | 3-8cm |
| SP22_BLS | Belixe Sul | Primary | Lower Jurassic | Nodule | Sporadic | Homogeneous | 3-8cm |
| SP23_BLN | Belixe Norte | Secondary | N/A | - | Abundant | Variable | - |
| SP24_CSV | Cabo S. Vicente | Primary | Lower Jurassic | Nodule | Sporadic | Homogeneous | 5cm |
| SP25_CSV | Cabo S. Vicente | Primary | Lower Jurassic | Nodule | Rare | Homogeneous | - |
| SP26_CSV | Cabo S. Vicente | Secondary | Lower Jurassic | - | Abundant | Homogeneous | - |
| SP27_CSV | Cabo S. Vicente | Primary | Lower Jurassic | Nodule | Rare | Homogeneous | 5-10cm |
| SP28_ASP | Aspa | Primary | Lower Jurassic | Nodule | Sporadic | Variable | 5cm |
| SP29_ASP | Aspa | Secondary | Lower Jurassic | - | Abundant | Variable | <5cm |
| SP30_FZF | Foz dos Fornos | Primary | Lower Jurassic | Nodule | Rare | Homogeneous | Max. 15cm |
| SP31_FZF | Foz dos Fornos | Sub-primary | Lower Jurassic | Nodule | Abundant | Variable | <5cm |
| SP32_FZF | Foz dos Fornos | Primary | Lower Jurassic | Nodule | Abundant | Variable | 4-8cm |
| SP33_FZF | Foz dos Fornos | Primary | Lower Jurassic | Nodule | Abundant | Variable | 4-8cm |
| SP34_PdA | Ponta dos Altos Este | Primary | Lower Jurassic | Nodule | Abundant | Variable | 4-8cm |
| SP35_BLX | Belixe | Isolated find | N/A | - | - | - | - |
| SP36_PtA | Ponta da Atalaia | Isolated find | N/A | - | - | - | - |
| SP37_PtA | Ponta da Atalaia | Secondary | Upper Jurassic | - | Abundant | Variable | - |
| SP39_AND | Andorinha | Secondary | Upper Jurassic | - | Sporadic | Homogeneous | - |
| SP40_FER | Ferrel | Primary | Lower Jurassic | Nodule | Abundant | Variable | - |
| SP41_FER | Ferrel | Secondary | Lower Jurassic | - | Sporadic | Variable | - |
| SP47_CdC | Casal da Colina | Primary | Middle Jurassic | Nodule | Rare | Homogeneous | 3-8cm |
| SP48_CdC | Casal da Colina | Secondary | Middle Jurassic | - | Rare | Variable | - |
| SP49_CdC | Casal da Colina | Secondary | Middle Jurassic | - | Rare | Variable | - |
| SP50_CdC | Casal da Colina | Primary | Middle Jurassic | Nodule | Sporadic | Homogeneous | 3-5cm |
| SP52_CdC | Casal da Colina | Secondary | Middle Jurassic | - | Rare | Homogeneous | - |
| SP53_GUI | Guilhim | Secondary | Middle Jurassic | - | Sporadic | Variable | - |
| SP54_GUI | Guilhim | Isolated find | N/A | - | - | - | - |
| SP55_CAL | Caliços | Secondary | Middle Jurassic | - | Rare | Homogeneous | - |
| SP56_JOR | Jordana | Sub-primary | Upper Jurassic | Nodule | Abundant | Homogeneous | 2-20cm |
| SP57_JOR | Jordana | Primary | Upper Jurassic | Nodule | Abundant | Homogeneous | 2-5cm |
| SP58_JOR | Jordana | Primary | Upper Jurassic | Nodule | Sporadic | Homogeneous | 5cm |

(*Continued*)

**Table 2.** (Continued)

| Sample ID | Outcrop name | State | Age | Chert morphology | Chert frequence | Chert variability | Chert size |
|-----------|--------------|-------|-----|------------------|-----------------|-------------------|------------|
| SP59_JOR | Jordana | Primary | Upper Jurassic | Nodule | Abundant | Homogeneous | 3-5cm |
| SP61_PER | Peral | Other | Upper Jurassic | Nodule | Sporadic | Homogeneous | 2-5cm |
| SP62_OLV | Oliveiras | Secondary | Middle Jurassic | - | Abundant | Variable | - |
| SP63_OLV | Oliveiras | Primary | Middle Jurassic | Nodule | Sporadic | Homogeneous | 2-15cm |
| SP64_OLV | Oliveiras | Isolated find | N/A | - | - | - | - |
| SP65_MALH | Malhão | Primary | Middle Jurassic | Nodule | Sporadic | Homogeneous | 5cm |
| SP66_MALH | Malhão | Primary | Middle Jurassic | Nodule | Sporadic | Homogeneous | 5cm |
| SP67_DESCAM | Cabeço Descampado | Secondary | N/A | Nodule | Sporadic | Homogeneous | - |
| SP68_PEDR | Pedreira | Other | N/A | - | Rare | Homogeneous | - |
| SP69_MAR | Praia da Mareta | Sub-primary | Upper Jurassic | Nodule | Abundant | Variable | Max. 25cm |

The age was defined based on the location of the outcrops, taking into consideration previous research work and geological maps. Samples which were isolated or in untrackable secondary deposition settings do not have a defined geological age.

Despite the similar characteristics between these cherts, the outcrops are heterogeneous and show varying characteristics between them, which may be of importance to distinguish between chert sources within the Lower Jurassic formation. These outcrops have been divided into four groups, following the available literature: 1) Cabo de S. Vicente (CSV) and Aspa (ASP); 2) Foz dos Fornos (FZF) and Ponta dos Altos (PdA); 3) Praia do Belixe (PBX, which includes Belixe Sul, BLS), and 4) Ferrel (FER). The Cabo de S. Vicente (CSV) and Aspa (ASP) chert outcrops are characterized by abundant nodules in the natural rock banks of the cliffs, appearing as horizontal layers within the parent rock. The banks seem to be mainly dolomite or dolomitic limestones. The process of dolomitization seems to have affected the chert nodules, as they often present different levels of silicification from the peripheral areas of the nodule to the interior, which also affects the size and feel of the grain. In this case, the peripheral areas of the chert nodules are more dolomitized, with visible grain and distinctively rough to the touch, while the interior areas are more silicified and conversely finer and smoother. The nodules vary in size, ranging from small 4 cm in diameter circular nodules to bed-like groups of nodules with ~20 cm in width. At the Aspa outcrops, the nodules are less frequent and smaller. Due to the proximity to the cliffs, the visibility of the chert nodules is good, and in present times, small chunks of chert (without cortex or with small amounts of parent rock attached) accumulate in secondary deposits nearby. Foz dos Fornos (FZF) and Ponta dos Altos (PdA) show similarities to the CSV outcrops. The nodules are visible in several banks of dolomite, dolomitic limestone, and limestone, partially covered by soil and sand. The nodules can be circular, around 5 cm in diameter, or wide nearly 20 cm in width. Despite their size, these cherts are frequently filled with fractures that fragment the larger nodules into smaller volumes of raw material. Alike CSV, FZF and PdA also show cherts with differing degrees of dolomitization, although in apparent smaller quantities than CSV. Besides the abundant presence of primary outcrops, there are also abundant chert nodule fragments in secondary deposition, down the slope of the cliff (in the case of FZF) or at the top of the cliff, on a sand path (in the case of PdA). These are small, between 1–4 cm in width, but of easy access. Between the FZF chert and the PdA, the main differences seem to be the cortex and parent rock, which show differing reactions to hydrochloric acid, the first being dolomite or dolomitic limestone, and the second being mostly limestone, with some degree of dolomitization in certain areas. Praia do Belixe (PBX) is characterized by the abundance of chert nodules throughout the dolomite layers of the cliff area. They are visible in certain areas of the cliff and within the rock shelters.

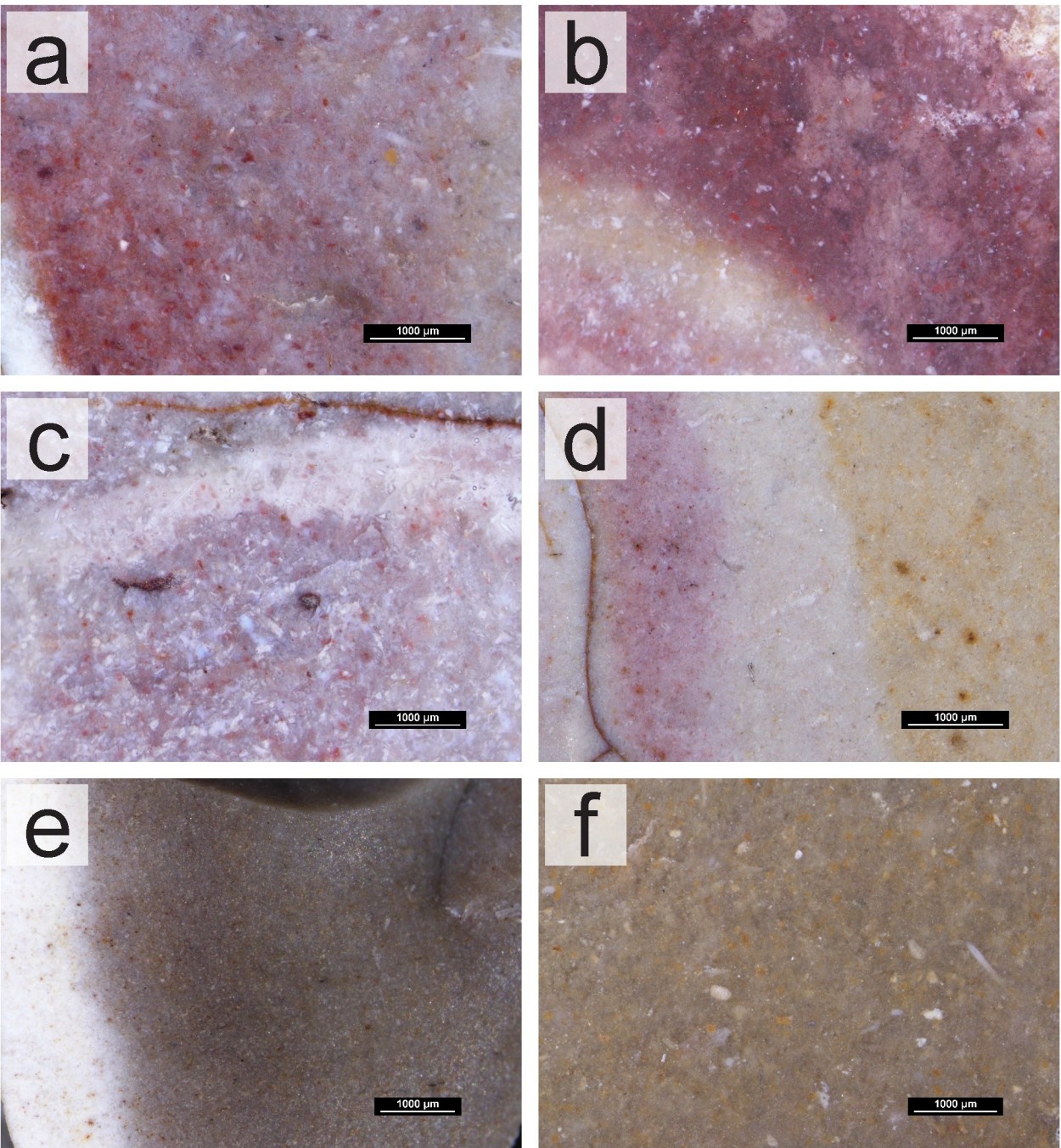

**Fig 4. View of the macroscopic variability of the Lower Jurassic Carixian cherts (several outcrops) from western Algarve.** (a) Sample SP34_PdA. (b) Sample SP14_PBX. (c) Sample SP16_PBX. (d) Sample SP30_FZF. (e) SP9_PdA. (f) Sample SP34_PdA.

The nodules can be small, around 5 cm in diameter, sometimes reaching more than ~30 cm in width, or bedded, as chert layers between the dolomite layers. The cherts show varying degrees of dolomitization and are mostly characterized by a coarse to semi-smooth feel and dull luster, often showing fractures and alterations. Unlike the other outcrops, no chert nodule fragments were found close to the cliffs, and the samples could only be recovered directly from the

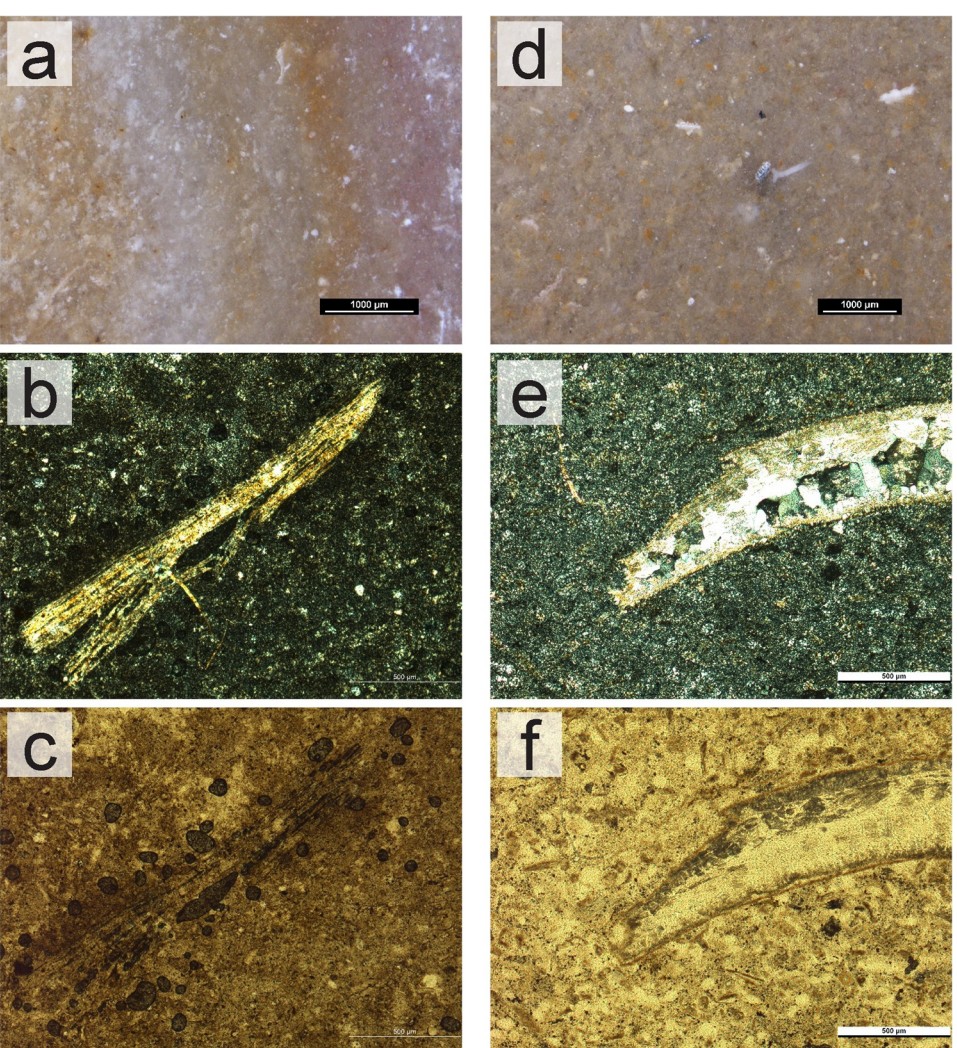

**Fig 5. Macroscopic and microscopic view of chert samples from the Lower Jurassic cherts from western Algarve.** (a) Sample SP7_CSV. Macroscopic view with a stereomicroscope. (b-c) Sample SP7_CSV. Microscopic view of thin section, XPL (b) and PPL (c). An Echinoid spine, replaced by quartz but preserving the fossil's original structure is present in the center of the images. (d) Sample SP34_PdA. Macroscopic view with a stereomicroscope. (e-f) Sample SP34_PdA. Microscopic view of thin section, XPL (e) and PPL (f). Detail of a bivalve shell replaced by at least two generations of quartz: microcrystalline quartz at the edges and macrocrystalline quartz inside of the shell.

embedded nodules in the cliff walls. Nodules scattered on the floor were only located at Belixe Sul (BLS), a primary outcrop nearly destroyed located on a field, north of the beach area. The chert in this outcrop showed no differences from PBX, aside from the size of the nodules, which were smaller and often showed signs of post-depositional alterations. A third location for chert has been previously identified north of BLS. Belixe Norte (BLN) is located on a dirt road and an unused agriculture field. Several chert fragments were collected in this location. However, BLN is in proximity to an archaeological site and several collected samples were lithic artifacts. No larger nodules or outcrop were identified in this location. The samples recovered from the location also seem to corroborate that BLN should not be considered an outcrop, as they do not match the local cherts and rather, resemble most of the samples recovered from eastern Algarve. Ferrel (FER), unlike the other outcrops, is located inland and away

from the coast. Due to its location in a homonymous village, the state of the outcrop is poor, and all samples were either recovered as scattered nodules or from larger blocks of rock, from a partially destroyed outcrop. The proximity of an archaeological site nearby also raises questions regarding the nodules found in secondary deposition, as these may be surface finds. Despite these caveats, the recovered samples are similar to those from the other Lower Jurassic outcrops, albeit with better quality, being characterized by a shiny to medium luster and smooth to semi-smooth feel. All surface fragments and nodules were small, with around 2 to 3 cm of width which may be explained by the state of the outcrop.

Contrasting with the diversity and quantity of the Lower Jurassic outcrops, there is only one identified outcrop for Upper Jurassic cherts in western Algarve, located at Praia da Mareta (MAR) and abundant, or in a secondary deposition at Ponta da Atalaia (PtA). The Upper Jurassic cherts are very similar to the Lower Jurassic, with dull to medium luster and grey/purple colors (Fig 6A). The translucency ranges from opaque to areas where the chert is translucent. This translucency may be a significant difference to distinguish between outcrops. Petrographically, the cherts are also similar to the Lower Jurassic ones. The only identifiable difference is the presence of calcispheres. All samples from the MAR and PtA outcrops seen under the petrographic microscope showed the presence of abundant calcispheres (Fig 6B and 6C), which is not always apparent with the stereomicroscope. Based on the presence of calcispheres, we may also consider the samples recovered at Andorinha (AND) to be Upper Jurassic (Fig 6E and 6F), which were uncommon and scattered at the top of the cliffs by the beach.

At Praia da Mareta the nodules are only easily accessible on the beach, where large boulders falling off the cliff (~1 m in diameter) are transported by the waves. Several chert nodules of different sizes can be found in the parent rock washed ashore, ranging between 2 cm to 20 cm in diameter. The quality of the chert also varies, possibly related to different dolomitization stages of the nodules, although this may also be influenced by chemical and physical alterations to the chert. At Ponta da Atalaia the chert can be found atop the cliffs, with rare nodules scattered on the floor.

## Eastern Algarve

On the eastern part of the Algarve, chert-bearing known formations are from the Middle and Upper Jurassic, known as the Malhão formation and the Jordana formation, respectively. The Malhão formation chert (Middle Jurassic) was identified in three outcrops: 1) Casal da Colina (CdC); 2) Oliveiras (OLV); and 3) Malhão (MALH). Whenever in a primary outcrop, this chert was homogeneous. Secondary deposits were also identified—Casal da Colina (CdC), Guilhim (GUI), Caliços (CAL) and Oliveiras (OLV) and were located in recent waterlines and slope deposits, and the cherts were often characterized by intense post-depositional alterations (in these cases, it was not possible to confirm the outcrop location). In the Malhão formation outcrops, the nodule frequency varied from common to abundant. The nodules are roundish, ranging between 3 to 5 cm in maximum width. In all cases, access to the outcrops was easy. Although the parent rock was hard, several chert nodules could be collected from the surface, accumulating further down in gentle slope deposits. The Malhão cherts show two differing macroscopic characteristics: pink/reddish/light gray cherts (Fig 7B) and grey cherts (Fig 7A). In general, they are both characterized by a dull to medium luster, opaque to sub-translucent translucency, and smooth to semi-smooth feel. They are easily identifiable through the high amounts of macroscopically visible inclusions, which look like white speckling in plain sight. Under the stereomicroscope, several round fossils and long spicule-like shapes can be identified. The petrographic analysis shows that the Malhão formation cherts are characterized by a wackestone texture, and composed of microcrystalline quartz (85–95%). Dolomite is also

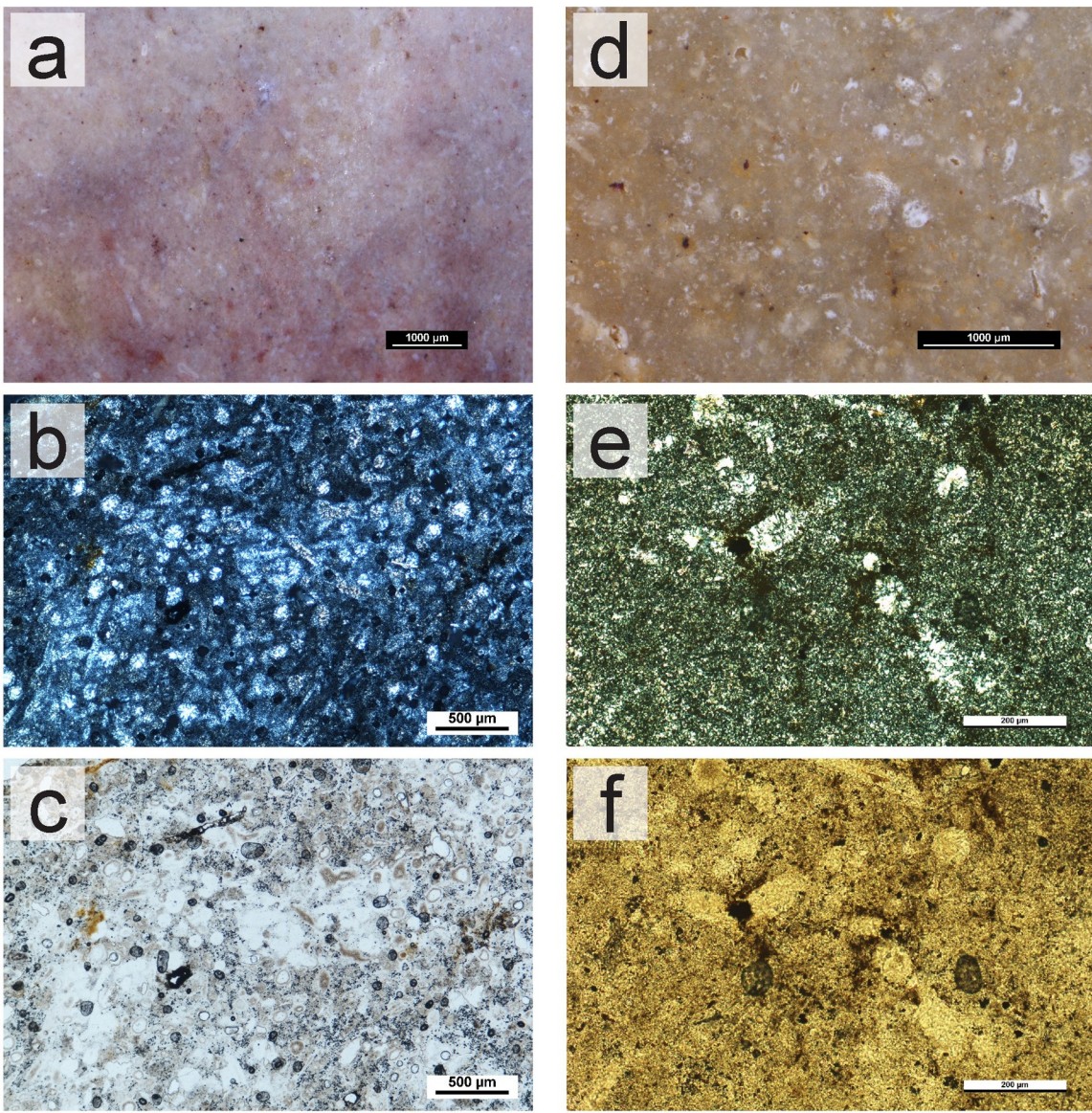

**Fig 6. Macroscopic and microscopic view of chert samples from the Upper Jurassic cherts from western Algarve.** (a) Sample SP36_PtA. Macroscopic view with a stereomicroscope. (b-c) Sample SP36_PtA. Microscopic view of thin section, XPL (b) and PPL (c). Several unidentifiable fossils can be seen in the photo, along with calcispheres. (d) Sample SP39_AND. Macroscopic view with a stereomicroscope. (e-f) Sample SP39_AND. Microscopic view of thin section, XPL (e) and PPL (f). A small amount of calcispheres is present in the image, along with few unidentifiable fossils replaced by quartz/chalcedony.

present (10–5%), as well as chalcedony and macrocrystalline quartz (<5%) frequently replacing fossils. Identified allochems are oxide patina, ranging from very frequent to uncommon. A high variety of identifiable fossils (although all are poorly preserved (Fig 7) were also identified. These fossils are Sponge spicules (Fig 7E and 7F), Radiolarians, Ostracods (Fig 7C and 7D), Echinoderms, Calcispheres, and possibly Tentaculites. Porosity in the samples occurs in small frequencies (<5%), of vuggy type.

The Jordana formation chert (Upper Jurassic) was identified in one area in the Algarve (Fig 3), in an outcrop of the same name (JOR). Whenever in a primary outcrop, the chert was homogeneous, although alternated with nodules of other lithologies within the parent rock.

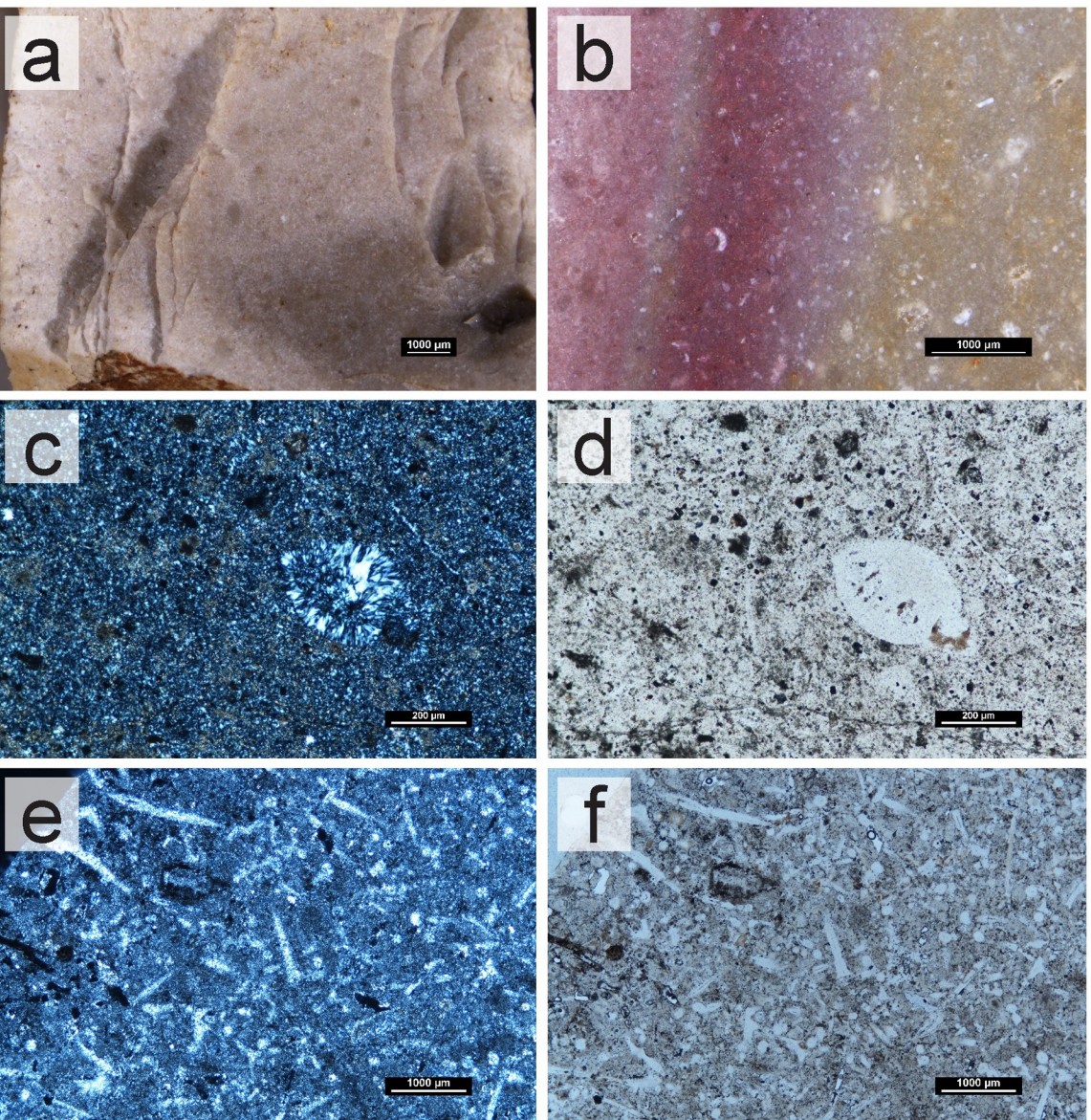

**Fig 7. Macroscopic and microscopic views of Middle Jurassic chert samples from the Malhão formation.** (a) Sample SP50_CdC, macroscopic view with a stereomicroscope. (b) Sample SP62_OLV, macroscopic view with a stereomicroscope. (c-d) Microscopic view of SP50_CdC, XPL (c) and PPL (d). Detail of a fossil (possibly an Ostracod), replaced by two generations of chalcedony (1st generation in the outer edges and 2nd generation replacing the inside). (e-f) Microscopic view of SP54_GUI, XPL (e) and PPL (f). General view of the thin section. Several fossil ghosts can be seen. Despite the poor preservation, it may be possible to identify a few fossils based on the size and morphology: 1) calcispheres or recrystallized radiolarians; 2) monaxon spicules pointed at one end.

No chert was identified in any secondary deposits, which might be related to the anthropic alteration of the landscape. Smaller nodules broken from the parent rock were identified near the primary source in a field. Whenever embedded in the parent rock, the nodules varied in size (~1–10 cm) and were abundant, with a high level of difficulty in their removal, due to the hardness of the parent rock. The cherts show little macroscopic variability between nodule and outcrop. They are grey/brown (with visible yellow inclusions) (Fig 8A and 8B). Within nodules, however, the cherts are heterogeneous, with dull and shiny or smooth and semi-smooth

feel areas. Some of the nodules also show variability of translucency, with translucent areas, with a very fine grain, and little presence of visible inclusions. The petrographic analysis shows that the cherts range from a wackestone to packstone texture (Fig 8), which was already seen macroscopically. They are composed mostly of microcrystalline quartz (90–99%), with the presence of fibrous chalcedony (1%) replacing the fossils and dolomite (1%), as well as negligible percentages of other minerals. Present allochems are iron oxides, ranging between very frequent to common. Albeit frequent, fossils are poorly preserved in general, with a few being identifiable: Calcispheres (Fig 8C and 8D), Bivalve shell (Fig 8E and 8F), Sponge spicules, Ostracod, Echinoderms, and Gastropod. Porosity is small (~1%) of vuggy type.

## Other outcrops

It is important to note that the aforementioned chert geological samples represent the chert variability of the identified chert-bearing formations and outcrops. However, a small number of outcrops described in regional geological maps were not identified or were not accessible. This includes four outcrops: 1) a Lower Jurassic outcrop with chert micronodules which was identified through a geological profile [84]; 2) a Middle Jurassic outcrop located at the easternmost section of the Algarve with partially dolomitized clasts and chert nodules [84]; 3) several unprospected locations from previous geoarchaeological works, all related to the Peral Anticline and reaching from central to eastern Algarve; 4) a possible chert exploitation archaeological site located on top of a chert source (Monte do Cerro), in eastern Algarve which was not identified and is possibly inaccessible [37]. All of these unidentifiable outcrops are located in the central or eastern portion of the Algarve. This may be related to the frequent landscape changes occurring due to agriculture or the population of previously uninhabited areas, which seldom occurs in western Algarve by the cliff areas, where most outcrops are located.

## Discussion

The survey work and analyses of the collected geological samples show that the south of Portugal has a high potential for chert raw material studies. The presence of chert-bearing geological formations throughout the Algarve would provide several possibilities for sourcing and procurement whenever groups moved throughout the territory. This is further important when we consider the geology of this region. The geology of the Algarve itself may have played an important part in how groups procured their raw materials, specifically, their chert, a task that has been identified as essential for hunter-gatherer groups. To the south, communities would only have access to chert-bearing outcrops down to the coast. To the north, the mountain range would not only have provided no chert nodules but may have also hampered the movement of populations, forcing groups to move east and west instead of north or south. This movement may have facilitated the gathering of cherts from different formations within the Algarve, posteriorly then brought into the sites. Especially for Middle and Upper Paleolithic occupations, understanding the sources of chert in the Algarve may provide data about where in the territory these groups were sourcing their chert raw materials, and how they were using the territory having in consideration the region's natural barriers and consequent distribution of resources. Although this topic remains unexplored, this study stands as a further step to tackle these questions in the Algarve, as it may provide the necessary basis for comparative studies with archaeological assemblages. However, tracking these movements and procurement patterns is only possible if the cherts from the different formations and outcrops can be traced back to their sources. This presented itself as the first caveat for this type of study, since for the Algarve, for example, all cherts formed in Jurassic formations in pelagic environments. Despite the similar formation environments, in general, there seem to be relevant differences

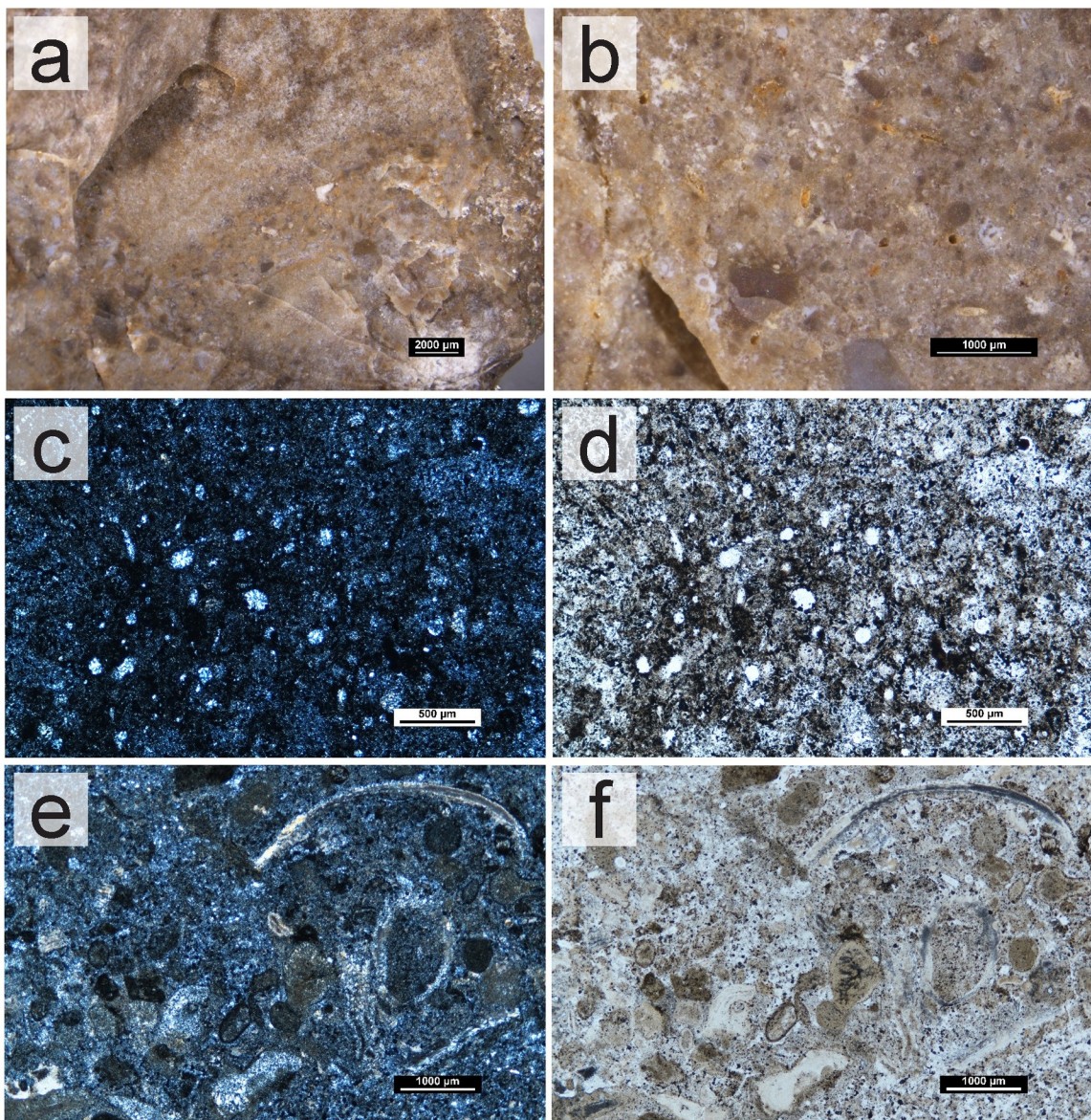

**Fig 8. Macroscopic and microscopic views of the Upper Jurassic chert samples from the Jordana formation.** (a) Sample SP58_JOR, macroscopic view with a stereomicroscope. (b) Sample SP59_JOR, macroscopic view with a stereomicroscope. (c-d) Microscopic view of SP56_JOR, XPL (c) and PPL (d). General view of calcispheres in an area of the chert characterized by a large concentration of oxide patina and opaques. (e-f) Microscopic view of SP58_JOR, XPL (e) and PPL (f). General view of the thin section. A large bivalve shell fragment is visible at the top.

between the cherts of different formations and periods. This is further relevant given the fact that they are geographically distant. Within formations, however, there are no discernable differences, both at a macroscopic and petrographic level, as these do not seem to be useful to distinguish between outcrops. This is most obvious on the Lower Jurassic formation of western Algarve. The identified chert groups, which varied mostly in color and fossil content, are present in several outcrops from this formation. In this region, the variables which may be better used to understand which outcrops were visited may be the quality (differing levels of dolomitization, presence/absence of fractures or even size of grain) and size of the nodules. The latter,

for example, is an important variable in the Praia do Belixe outcrops, which show the largest volumes of rock, even if the chert's quality is worse than some other available, smaller nodules. Size may be used in conjunction with other technological data, to understand whether different nodules were being explored differently based on their size, or their procurement was being preferred in relation to other smaller nodules in possibly closer outcrops in the region. The Upper Jurassic nodules of western Algarve also show larger volumes than those from the Lower Jurassic. Translucency also seems to be a good macroscopic indicator to distinguish between western Algarve Lower and Upper Jurassic cherts, since the latter are characterized as sub-translucent. However, for a reliable distinction between the Lower Jurassic and the Upper Jurassic cherts, petrographic analyses and the identification of calcispheres may be necessary. The differences identified among the cherts of the various formations can be seen both at a macroscopic and petrographic level. Given the formation settings, petrographically, all the cherts from the Algarve are fairly homogeneous - marine origin, in limestone or dolomitic limestone rocks, all formed during the Jurassic. The use of specific fossils for the identification of the cherts is also difficult since these are often not well preserved enough to allow the identification of species that may connect a group of cherts. The size, frequency, and preservation state of the fossils seem to be, then, one of the defining criteria for discerning cherts from different formations, and thus, different geographic areas. These characteristics seem to be observable macroscopically, as well, allowing the cherts from the three different areas and formations–West (including the Lower Jurassic and Upper Jurassic formation cherts of western Algarve), Jordana, and Malhão - to be differentiated without the need for thin sections (Fig 9). An exception might be the distinction between the cherts from the West and Malhão - the reddish cherts from Malhão are visually indistinguishable from those from the West with a higher fossil content. The grey cherts of the Malhão Middle Jurassic formation do show a higher concentration of visible round fossils, however the distinction is only possible seen under the stereomicroscope and on a fresh surface, which might hamper the classification of archaeological materials. These distinctions are especially important for archaeological collections, especially those which may be small, with small artifacts, or for the study of older collections to which other (destructive) means of analysis may not be applied.

These data seem to confirm the potential of a macroscopic analysis to study the cherts of the Algarve. Albeit applying different methodologies, such as petrographic analyses, to these cherts is a way of completing the petrographic study of a collection, reliably applying mostly a macroscopic analysis to the assemblages coming from southern Portugal helps tackle issues such as the destructiveness, costliness, and time consumption of some methods. Our study was able to provide a more detailed reference collection for chert outcrops in the Algarve, which will allow to test models about raw material procurement and use in a multilayered archaeological site like Vale Boi.

There are, however, some noteworthy caveats in this type of study. Landscapes have changed through time, both naturally and with the influence of modern society. House constructions, agricultural fields, and roads, for example, have modified the landscape, possibly altering the availability and visibility of raw materials. Other natural processes, such as the development of biomantle or soil cover may also hamper raw material visibility in the landscape. Similarly, environmental changes may have had an impact on raw material availability, through its impact on surface processes which expose, erode and transport the raw materials [101]. As such, it is important to keep in mind that current raw material sources, and specifically chert ones which may be subtly visible in the landscape, may not correspond to the sources which were available in the past. Another caveat regarding chert sources, especially in a geographic area like the Algarve, is the possibility of some outcrops being submerged. Previous studies have identified the existence of Jurassic lithologies on the west coast, submerged by

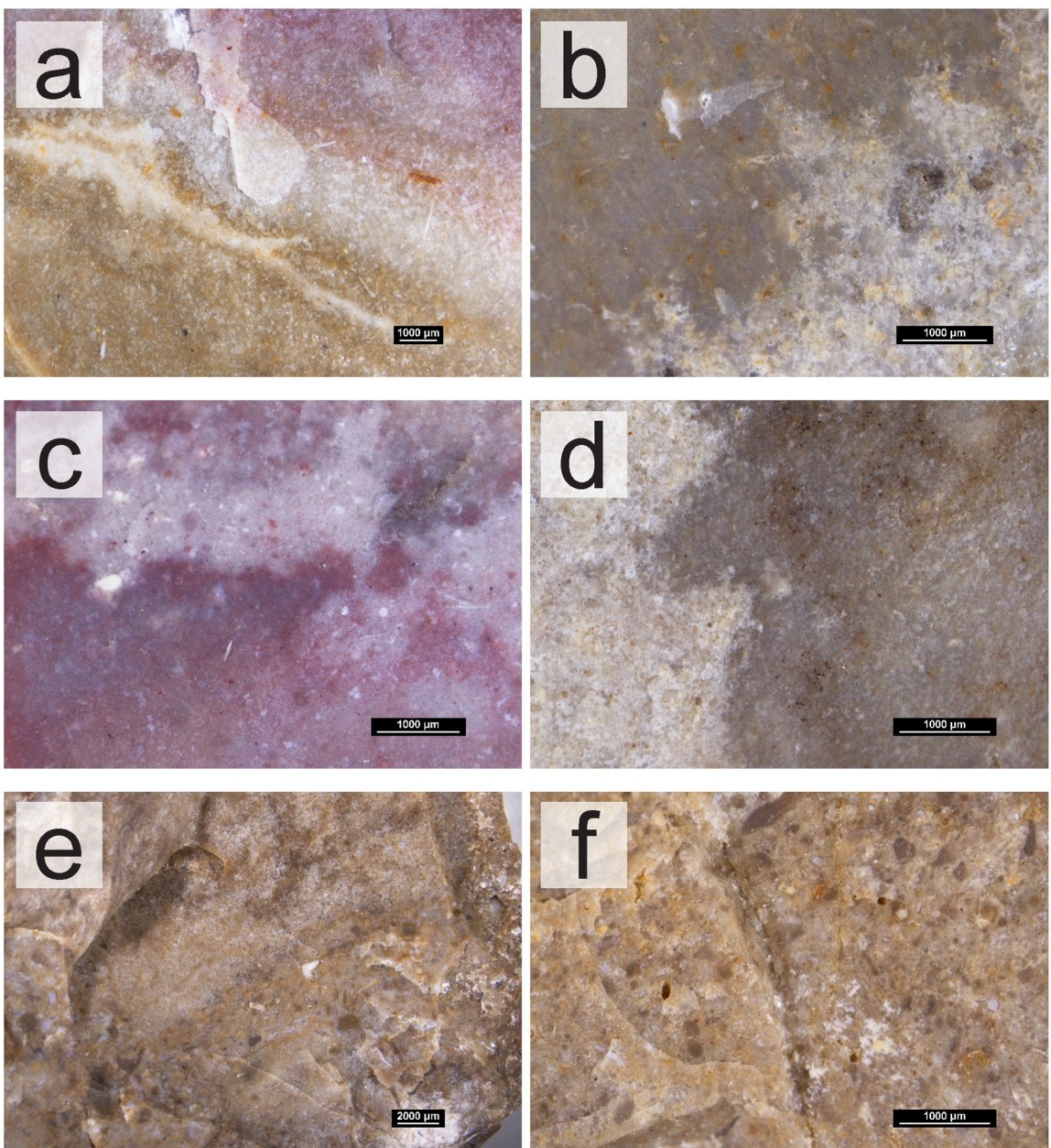

**Fig 9. Comparison between the cherts of different formations in the Algarve, organized by geological age and formation.** (a-b) Lower Jurassic (Carixian formation) chert samples under the stereomicroscope. (c-d) Middle Jurassic (Malhão formation) chert samples under the stereomicroscope. (e-f) Upper Jurassic (Jordana formation) chert samples under the stereomicroscope.

water [81]. These were mainly surveys done by oil companies which were able to obtain the submerged stratigraphies on the southwestern coast of Portugal and that revealed Jurassic limestones and dolomites, although the presence of chert is not described and remains unknown. Whether these submerged lithologies are different from the currently emerged ones is also uncertain. These studies also revealed that Jurassic lithologies were covered by more recent geological layers, including Pliocene and Pleistocene layers, thus forming before and/or during Paleolithic occupations of the territory [81]. In times when the sea level was similar to

the current one, the submerged lithologies would not have been accessible, even during low tide. However, during periods when the sea level was lower due to water freezing in the polar caps, as during the LGM for example, large portions of the coast that had been submersed would have been accessible. During this period, in Portugal, the coast would probably be close to the continental platform, ~120 m below the sea level [102, 103]. Specifically in western Algarve, the coastline may have been displaced ~10–15 km offshore from the present coastline [104]. Although currently unknown, it may be possible that during periods like the LGM, limestone and dolomite Jurassic lithologies, with chert nodules, might have been exposed and available. When studying chronologies characterized by cold and harsh climatic conditions with impacts on sea level changes and mixed with coastal uplift events, it is relevant to keep in mind that the chert variability present in the current coastline may not necessarily reflect the variability in the past. Despite these caveats, our data raises the possibility to understand whether this new portion of landmass altered the raw material procurement patterns of these groups, or added new resources which had been previously unavailable. Studies that compare Gravettian and Magdalenian assemblages (with higher mean sea-levels, and possibly even similar to current coastlines) to Proto-Solutrean and Solutrean assemblages (with LGM low mean sea-levels), within one single site, may give new insights into this question.

## Conclusion

In this study we identified several sources of chert nodules in southern Portugal and characterized the regional cherts, which were of critical importance for hunter-gatherer communities during the Late Pleistocene. For this we applied a two-step raw material analysis approach, composed of macroscopic and petrographic analyses. The results show the presence of four different chert formations, dispersed in western and eastern Algarve. Within most formations, there is variability in the nodules and the outcrops. There are however identifiable macroscopic and petrographic differences between formations which allow their distinction. Although the petrographic analysis is essential to identify the fossils present in the chert, a macroscopic approach seems to be pertinent for a quick and inexpensive analysis to distinguish between cherts of the different formations. The presence of chert sources in the Algarve region, with distinguishable characteristics between formations, which may be analysed preliminarily through macroscopic approaches, shows the potential for chert raw material studies of archaeological sites in this key area. Further steps in our study will include the use of the data gathered in the present study and the completed LusoLit lithotheque to study chert use from multi-component sites with Upper Paleolithic chronologies such as Vale Boi, and participate in the discussion of human adaptations throughout the Late Pleistocene. Furthermore, future approaches include the use of geochemical methods to further characterize these cherts and the integration of the resulting data in the online database.

## Supporting information

**S1 Table. Field sample/outcrop dataset.** Table with recorded data on the outcrops and cherts, during fieldwork and prospections during 2021 and 2022.
(XLSX)

**S2 Table. Macroscopic analysis dataset.** Table with the data collected from the macroscopic analyses of all chert samples collected from the fieldwork and prospections during 2021 and 2022.
(XLSX)

**S3 Table. Data dictionaries for macroscopic and petrographic analyses.** Description of the used variables (including allowed variables and references) for the macroscopic and petrographic analyses used in the present study.
(PDF)

## Acknowledgments

We would like the thank the entities that funded the current research: Fundação para a Ciência e a Tecnologia (FCT) and ICArEHB (Interdisciplinary Center for Archaeology and the Evolution of Human Behavior). Furthermore, we want to thank Dr. Telmo Pereira and Dr. Carlos Ribeiro for allowing us to consult their previous research and for their support. We'd like to thank Jack Acres for the IT support essential for the creation of the online LusoLit database. Finally, we thank Dr. Célia Gonçalves for helping with the creation of the maps.

## Author Contributions

**Conceptualization:** Joana Belmiro, Xavier Terradas, João Cascalheira.

**Funding acquisition:** Joana Belmiro, João Cascalheira.

**Investigation:** Joana Belmiro.

**Methodology:** Joana Belmiro, Xavier Terradas, João Cascalheira.

**Writing – original draft:** Joana Belmiro.

**Writing – review & editing:** Xavier Terradas, João Cascalheira.

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
