## [Decision Letter · Decision Letter 0]

30 Mar 2023

PONE-D-22-35505Creating frames of reference for chert exploitation during the Late Pleistocene in Southwesternmost Iberia.PLOS ONE

Dear Dr. Belmiro,

Thank you for submitting your manuscript to PLOS ONE. After careful consideration, we feel that it has merit but does not fully meet PLOS ONE’s publication criteria as it currently stands. Therefore, we invite you to submit a revised version of the manuscript that addresses the points raised during the review process.

We look forward to receiving your revised manuscript.

Kind regards,

Enza Elena Spinapolice, Ph.D

Academic Editor

PLOS ONE

Journal Requirements:

2. In your manuscript, please provide additional information regarding the specimens used in your study. Ensure that you have reported specimen numbers and complete repository information, including museum name and geographic location. 

For more information on PLOS ONE's requirements for paleontology and archeology research, see https://journals.plos.org/plosone/s/submission-guidelines#loc-paleontology-and-archaeology-research.

3. Please amend your manuscript to include your abstract after the title page.

4. We note that Figure 2 in your submission contain copyrighted image. All PLOS content is published under the Creative Commons Attribution License (CC BY 4.0), which means that the manuscript, images, and Supporting Information files will be freely available online, and any third party is permitted to access, download, copy, distribute, and use these materials in any way, even commercially, with proper attribution. For more information, see our copyright guidelines: http://journals.plos.org/plosone/s/licenses-and-copyright.

5. We note that Figures 1 and 3 in your submission contain map/satellite images which may be copyrighted. All PLOS content is published under the Creative Commons Attribution License (CC BY 4.0), which means that the manuscript, images, and Supporting Information files will be freely available online, and any third party is permitted to access, download, copy, distribute, and use these materials in any way, even commercially, with proper attribution. For these reasons, we cannot publish previously copyrighted maps or satellite images created using proprietary data, such as Google software (Google Maps, Street View, and Earth). For more information, see our copyright guidelines: http://journals.plos.org/plosone/s/licenses-and-copyright.

a. You may seek permission from the original copyright holder of Figures 1 and 3 to publish the content specifically under the CC BY 4.0 license.  

Reviewers' comments:

Reviewer's Responses to Questions

**Comments to the Author**

1. Is the manuscript technically sound, and do the data support the conclusions?

Reviewer #1: Yes

Reviewer #2: Partly

Reviewer #3: Yes

2. Has the statistical analysis been performed appropriately and rigorously? 

Reviewer #1: N/A

Reviewer #2: N/A

Reviewer #3: N/A

3. Have the authors made all data underlying the findings in their manuscript fully available?

Reviewer #1: Yes

Reviewer #2: Yes

Reviewer #3: Yes

4. Is the manuscript presented in an intelligible fashion and written in standard English?

Reviewer #1: Yes

Reviewer #2: Yes

Reviewer #3: Yes

5. Review Comments to the Author

Reviewer #1: Paper present an unedited work on raw material resources from southern Portugal. This work is well presented and introduced by reference to the scientific context. Methods are well explained and presented as well as their potential limits.

Reviewer #2: The manuscript led by Joana Belmiro entitled “Creating frames of reference for chert exploitation during the Late Pleistocene in Southwesternmost Iberia” are a type of works very interesting and important for the development of research on the territorial mobility of hunter-gatherer societies. In particular, this work focuses on southern Portugal, a region with important Upper Pleistocene archaeological settlements and with few detailed studies on possible source areas, in particular chert. Despite the great interest it arouses, the work reflects a number of limitations that need to be addressed in order to effectively help the development of the study of lithic raw materials and their catchment areas in this region.

Starting with general aspects, I will then go on to cite detailed aspects in the order of the manuscript.

If the aim of this paper is to create a good frame of reference for the identification of the different types of cherts in the archaeological assemblages by presenting the siliceous variability in the territory, first of all a geological map of the region has to be created, showing the siliceous outcrops in a clear way with their geological formations and with the outcrops individualised. General photographs of the outcrops would not be superfluous, as there is only one figure showing only one outcrop.

In relation to the nomenclature used to define each outcrop/variety of chert must be clear and consistent. It is not possible to talk about outcrops, chert varieties, geological age, indistinctly to ascribe each time to a group. Personally, I think it needs to be clearer.

Despite this work attempts to be a frame of reference for the chert source areas for southern Portugal, some archaeological context, I believe it is necessary.

In terms of structure, there are a number of shortcomings: in materials and methods, it must be clear what the starting point is, what materials are to be studied, and they must be well organised so that the presentation of data flows in a more organised way and following the same terminology. The results are divided into western, eastern and other outcrops, when this division does not make much sense and it would be better to divide them by geological periods. A more detailed petrographic description would also be needed in many cases. For example, presence of silica forms and percentages, matrix, replacements, porosities, autochthonous minerals...

The discussion should be contextualised with some archaeological sites, especially when talking about the important role of chert in the region. For example, line 570 states that all chert comes from the Jurassic formations. The only thing we note is that we only have chert from these geological layers in the region, but this does not mean that the chert found in the archaeological sites all comes from these formations. In the same way in line 623 comments that these studies will be very useful for the study of the lithic origin of the different archaeological levels of the Vale Boi site, but nothing is said about the archaeological assemblage, nor about the predominant raw material.

It is very interesting when we talk about the variability of chert in the same geological formation. This is a very common problem for all of us who study lithic provenance, so the attempt to find solutions is always welcome. In this case the identification is through the fossils content, which is not always possible. I think at this point the discussion could have been developed a little more. In addition, it contradicts itself (paragraph from 595 to 610). First saying that the fossils are hardly preserved and therefore this criterion cannot be used for their identification, and then that it is the only way to be able to solve the great homogeneity they present. This should be better clarified.

Finally, I fully agree that macroscopic as well as petrographic criteria are the best to establish provenance of lithic assemblages.

Specific comments:

In the abstract when you use the concept “use studies” you mean traceological studies? Better if you use this term or use-wear analysis.

Figure 1. Some more data is needed. Altimetry, name of geological formations, main geological units, some toponym. It will help to the lector to be able to situate when formations, geomorphological units, etc., are mentioned throughout the manuscript.

Line 180. There is an error when talking about Paleozoic sedimentation and it is mentioned that is starts during the Triassic. This also leads to a certain disorganisation when talking about the geological formations and sedimentary processes of the region as they are usually presented in chronological order, either by eras (Paleozoic, Mesozoic…) or systems (Triassic, Jurassic…). It doesn’t help much when talking about the Mesozoic and Cenozoic materials of the Algarve basin no reference about the Cenozoic materials are mentioned in the Geological settings. In general terms, the geological settings are somewhat confusing.

Lines 187 and 195. Paleozoic is mentioned again. Did you mean Mesozoic?

In subsection 2.2, it begins with the generalisation that the presence of chert is usually associated with carbonate formations of limestone and dolomite. This is not always the case, and even less in the Iberian Peninsula, where evaporite formations with chert nodules are abundant in the centre and north. I think this should be clarified, either by focusing on the study area or by mentioning the evaporite formations. In general, this subsection must be completed. The process of chert formation in these formations is not mentioned, is it related to the accumulation of siliceous skeletons on the seabed? How chert is formed? Also, all geological formations with cherts are not marked in figure 1. This will help to the lector.

Line 201. The error of lumping the Triassic and Jurassic formations into the Paleozoic continues.

Line 217. In the same way that chert formations can be associated with Middle and Upper Jurassic geological formations (Malhão and Jordana formations respectively), it is possible to know which geological formation carries the Lower Jurassic chert from the outcrops of Cabo S. Vicente and Praia do Belixe?

Line 277. Reference Soler et al 2020, must be changed by Gómez de Soler et al. 2020.

It is important to define which nomenclature you will use for mention the chert varieties: its outcrop name, ID sample, geological age, geological formation, variety and always use this nomenclature to refer to them. Is quite confusing.

Table 1. It would be good to add a column with the geological formation and another with the system or epoch. When talking about TSL - Thin Section Lab, to which laboratory do you refer? Specify.

Line 338. When citing figure 3, it would be better to divide it into 3a, 3b and 3c for quicker identification (this is also valid for the rest of the figures). In addition, it would be quite clarifying to be able to identify the dots with the outcrops, as throughout the presentation data figure 3 is quoted for outcrops but there is no way of knowing which ones it refers to. Also, the figures below need to have scale, as the only visible scale does not correspond to these figures.

Line 347. At the footer of table 2 I would remove the term “nodule” by “chert” as some of the outcrops chert’s morphology are not in the form of nodules, isn't that right? In the same table why the sub-primary ouctrops the morphology of the cherts is not given. In this case they would be blocks, wouldn't they?

Line 357. When talking about 3 varieties of Lower Jurassic cherts and you said that are grouped through color and fossil content, it would be easier for the reader if the colons and semicolons were followed by 1); 2) and 3).

Line 369. At the footer of figure 4 you need to add to figure 4a “Sample”.

Paragraph from line 373 to line 380 is confusing. If at the beginning you are talking about MCF variety and then for the microscopic analysis you are talking for all Lower Jurassic cherts you must add a full stop and specify that more than 50% of the samples that are not from, the MCF variety fossils are difficult to identify. If, on the other hand, you are always talking about the MCF variety, there is a major contradiction, when it is a variety with fossils content a naked eye, but then you affirm that more than 50% of the samples do not show them, then they are other varieties or this variety is poorly defined. However, at the end of the paragraph, when you quote figure 5, it corresponds to a sample of the MCF variety ¿no? No way to know it because in the footer of the figure 5 only de sample ID appears and no the type of chert variety (it is strongly recommended to add it, in the same way that in table 2).

Continuing with this confusion, from line 390 onwards, we no longer refer to varieties of cherts or geological ages, but chert outcrops to be divide them again into groups. It would help if it could be known which varieties of chert occur in which outcrops. It would help also some figures with the different types of chert outcrops and the location of these outcrops.

Line 392. The outcrops are said to have been divided into four groups but then five groups are cited: 1) Cabo de S. Vicente, 2) Foz dos Fornos, 3) Ponta dos Altos, 4) Praia do Belixe, and 5) Ferrel.

At the beginning of line 390 it is said: “Despite the similar characteristics between these cherts, the outcrops are heterogeneous and show varying characteristics between them, […]” but then when talking about FZF and PdA outcrops it is said that shows similarities to CSV ouctrop. Then why this division?

Line 415. Better refer to “secondary deposits” no “secondary deposition”. In this case, due to their proximity to the primary outcrop and, I imagine little roundness, it could be considered as a sub-primary deposit.

Line 447. When changing the subject from Lower Jurassic to Upper Jurassic outcrop types, it would be good to put a full stop. At this point it would be advisable to follow the same presentation scheme. If for the Lower Jurassic first the chert varieties are presented and then the outcrops, why for the Upper Jurassic first the outcrop is presented and then the macroscopic and petrographic characteristics.

Line 475. Subsection 4.2. It is important to present data in a clear, concise and homogeneous way. When discussing the Middle Jurassic cherts of the Malhão Formation, it would be good to cite first the localised outcrops and their locations throughout the text depending on what is to be explained at any given moment. It does not help to follow the reading in an easy way.

In figure 7, subfigure 7b cites an outcrop of such formation that has not been cited anywhere in the text.

Line 534. Subsection 4.3. Other outcrops are mentioned but their names are never mentioned and their locations cited. If the intention is to create a frame of reference for the scientific community on possible chert source areas, the presentation of the data needs to be clearer.

Line 551. At the beginning of the discussion it is commented that through the surveys it has been observed the enormous potential that the south of Portugal has for the study of lithic raw materials, especially chert. Readers are not aware of this, as only this raw material is presented in the manuscript, so this sentence should be qualified.

Line 552. In the results section as well as in the figures are organised in the west and east zone, it does not make much sense that now in the discussion let us mention the central zone of the south of Portugal.

Line 602. Figure 9, you must put a white background.

Line 668. Four different chert formations. Which ones? In the abstract is talking about three and in the manuscript are only two mentioned: Malhão Formation from Middle Jurassic and Jordan formation from Upper Jurassic. Formations, geological ages and outcrops are confused throughout the manuscript. This causes confusion.

In the references section, I believe that the authors do not follow the journal's citation form. In all of them the year is after the authors, the names of the journals are not abbreviated, nor is the type of separation between volume and number of pages. There are also some errors in the citation, for example:

Line 798. In Flügel's reference, the name of the book is missing (Microfacies of Carbonate Rocks).

Line 808. In Gómez & Lunt reference, the name of the book is missing (Phylogeography of Southern European Refugia)

Line 884. Incomplete reference.

Line 928. Incorrect reference, change Soler B.G by Gómez de Soler, B.

In supplementary materials, it is not clear to me that in the "outcrops_FULL_db" document the exact coordinates of each outcrop have to be given. We are all aware of the erosion of chert outcrops and the bad practices that occur in them. I think that with an approximate coordinate or quoting a geographical area would be enough. If there is someone with a scientific interest in knowing the coordinates, there is always a way to get them to you. This is just a personal opinion.

Reviewer #3: As it is mentioned in the title, the paper deals with the creation of a databse about chert types in the southernmost region in Portugal, and this for improving raw material studies of Late Pleistocene archaeological sites. The structure of the manuscript is clear and logical. The data presented in SOM are rich in information, The published Analysis dataset is very welcome not only for understanding the presented data structure and conclusions, but also for creating similar databases in other regions or countries. The description of the used methodology is complete and useful for those researchers who plan to make such kind of investigations. Data concerning the abundance and size and morphology of chert blocks at the source are highly important from an archaeological point of view regarding studies about raw material economy.

The text seems for me easily readable and understandable but I can not evaluate the level of English because I am not a native speaker.

I suggest minor revision concerning the following points.

When arguing in favour of creating reference databases of raw materials (line 125-128), the authors cite three cases related to an archaeological site, as well as the LIR of Ireland. However, there are important lithotheques in Iberia too (i.e. LITHICUB and LITOCAT in Barcelona). Moreover, the most important references in this field are the lithotheque projects in France (works and publications by V. Delvigne, J. Féblot-Augustins, P. Fernandes, A. Morala, A. Turq and others), where these researches started as early as in the 1980s.

As it is expressed, the main goal of the present study is to establish a reference for cherts in an understudied region, integrated into the LusoLit lithotheque (line 145-152); the geological samples studied came from fieldwork in 2021-2022, and the prospected locations were chosen after reviewing previously known research (line 253-261). However, it seems that the sampled locations presented in Fig. 3 already existed in LusoLit, as far as the Figure 1 of Pereira et al. 2016b (cited in the manuscript) can show it. For this reason, it would be welcome to read more about how the present study completed or enriched the former data of LusoLit (see line 261-263).

Besides macroscopic and petrographic (thin sections) analyses, the multilayered approach in raw material studies, as referred by Brandl 2016 in line 245, contains geochemical analysis too. A more complete description of this methodology was published by M. Brandl in Archaeologia Austriaca 97-98 (2013-2014) p. 33-58. Do the authors expect to make geochemical analyses too? If so, in what sens these analyses can improve the distinguishing between sources. If not, why.

The list of References needs to be checked thoroughly. There are typing mistakes, typographical faults (lower case/upper case), lacking bibliographical data.

Finaly, some corrections to verify.

line 32: Southwestern instead of Southernwestern

line 33: Cascalheira et al. 2017b appears first here, Cascalheira et al. 2017a only in line 276, change 2017b and 2017a along the manuscript

line 53-57: add a reference for Vale Comprido point and its association with the Heinrich Event 2 episode

line 180-201: Triassic and Jurassic formations belong to Mesozoic and not to Paleozoic

line 182: western and eastern (not oriental) sub-basins

6. PLOS authors have the option to publish the peer review history of their article (what does this mean?). If published, this will include your full peer review and any attached files.

Reviewer #1: No

Reviewer #2: No

Reviewer #3: No

---

## [Author Response · Author response to Decision Letter 0]

9 Jun 2023

Reviewer #1:

We appreciate the comments made by Reviewer #1.

Reviewer #2:

The authors would like to thank reviewer #2 for the diligent comments, questions and corrections to the manuscript. We believe these were beneficial and had a positive impact on the scientific quality of the paper.

Added more photos of the outcrops. A geological map with the siliceous outcrops, recovered samples, and geological formations has been created.

The authors have revised the text to make the nomenclature clearer and more consistent.

The authors provide a contextualization of the importance of chert in the archaeological record, in order to provide an archaeological context for the work. Since future research will focus on the use of chert in a specific site (Vale Boi), but the current research and results do not need to be limited to one site or chronology, we decided a general contextualization would be enough. 

The results are divided between western and eastern since during the period where these cherts formed, the Algarve was divided in a west sub-basin and an east sub-basin. These sub-basins did not always have the same conditions—it seems important for the authors to respect this division as it provided clear differences in the formations and cherts. We have also homogenized the nomenclature and made the text clearer whenever needed. We have completed the petrographic descriptions to include pertinent, more detailed data for the characterization.

Regarding the first bit, that is what the authors meant—all the chert from the geological layers in the Algarve comes from Jurassic formations and pelagic environments. We exchanged the term “(…) found in Jurassic formations” for “(…) formed in Jurassic formations” to make it clearer we are only talking about geology and not archaeology. Added a reference to the predominance of chert in the lithic assemblage of Vale Boi for better context.

What the authors mean is that fossils in the cherts do not seem to show preservation enough to clearly identify a fossil species, and thus fossil species are not ideal for distinguishing the cherts from the Algarve. Instead, a combination of size, frequency, and even preservation seem to be the defining criteria to discern between the cherts. In the near future, taking a larger number of samples may provide more data on the associations of microorganisms that may be useful for a better characterization of the chert’s formation environments. Despite this, we made a slight alteration to the text to make our point clearer.

By chert procurement and use studies we mean studies that focus on how past hunter-gatherers obtain and use chert. To make it clearer, we replace “use” with management: “(…) chert procurement and management studies in this region (…)”.

Main geological units and formations have been better represented in a new map, which also includes reference points (toponyms). The altimetry is present in the second map.

Paleozoic has been replaced by Mesozoic. Likewise, the geological settings are described now mostly using systems.

"Lines 187 and 195. Paleozoic is mentioned again. Did you mean Mesozoic?" Solved throughout the manuscript.

Clarified and completed the paragraph on the formation of chert in the Algarve.

In the geological literature, the Lower Jurassic cherts from outcrops like Cabo S. Vicente are not associated with named formations like those from Malhão or Jordana. Rather, they are called Carixian formations - Limestones and dolomitic limestones with chert nodules. The same can be said for the Upper Jurassic chert of the Atalaia area (Praia da Mareta), where the naming of the formation may simply be Kimeridgian formation. We have added reference to these formations (Carixian and Kimmerigian) in the manuscript, to make it clearer and more homogeneous.

Reference "Gómez de Soler et al. 2020" has been corrected.

Solved. For the cherts of the Lower Jurassic (Carixian) formation, we use the chert variety nomenclature for a clear macroscopic characterization. Every other case, we use the geological formation to refer to the cherts, and name specific outcrops whenever they are being described or there are particularities that need to be individualized and specifically explained.

1) Columns with the geological formation and epoch were added to Table 1; 2) TSL – Thin Section Lab is the actual name of the lab, located in France. To make it clear TSL is a specific laboratory and not a generic reference, in the legend we added within brackets the city and country of the lab. To homogenize the legend, we added the University of Barcelona’s lab name, and city name.

All figures have been renumbered for quicker identification (ex. Fig. 7 is now divided into a to f). Scales have been added to the detailed views of the outcrop clusters. Individual numbers have been given to the points on the map. In the legend, we specify which points constitute which outcrops, so it is possible to understand where the outcrops are on the map.

Term nodule at the footer of table 2 has been replaced.

Added 1), 2) and 3) to Line 357.

"Line 369. At the footer of figure 4 you need to add to figure 4a “Sample”." Solved.

A full stop was added to make the distinction clearer. When describing the petrography, we mean the sum of all Lower Jurassic cherts. Figure 5 does not show a sample of MCF, but was chosen rather to show some of the identified fossils within the samples (since as mentioned, fossils are frequently poorly preserved).

The authors have reviewed the text in order to homogenize the terms which are used, to make the results and discussion clearer and easier to follow. Chert variety is used for the Lower Jurassic cherts to describe them in an organized way since there is plenty of variety. Geological ages and formation names (whenever these names exist, as in the case of Malhão and Jordana) are used throughout to discern between the cherts from different formations. Outcrops are used whenever there are relevant differences within the formations.

Line 391. Solved.

The division between outcrops in western Algarve is following previous geological works in western Algarve, as mentioned in the text. They represent outcrops and geological cuts in different areas throughout the formation. Since this division is prevalent in the literature, the authors decided to maintain it.

Solved. The authors replaced the term "secondary deposition". We maintain the secondary deposits, since the top area of Aspa does not show just a little roundness, but rather small and very rounded chert blocks.

"Line 447. When changing the subject from Lower Jurassic to Upper Jurassic outcrop types, it would be good to put a full stop. At this point it would be advisable to follow the same presentation scheme. If for the Lower Jurassic first the chert varieties are presented and then the outcrops, why for the Upper Jurassic first the outcrop is presented and then the macroscopic and petrographic characteristics." Solved.

"Line 475. Subsection 4.2. It is important to present data in a clear, concise and homogeneous way. When discussing the Middle Jurassic cherts of the Malhão Formation, it would be good to cite first the localised outcrops and their locations throughout the text depending on what is to be explained at any given moment. It does not help to follow the reading in an easy way." Solved and extended to the Jordana section for homogeneity.

Figure 7b refers to sample SP62_OLV. OLV (Oliveiras) refers to the name of the outcrop, which belongs to the Malhão formation according to the geological maps. This sample appears in table 2.

Line 534. These other outcrops often lack a specific location, or specific name or are just short mentions in other works or sources. The authors decided that to make the reading more streamlined in this section, it would be enough for the outcrops to be mentioned (using the defining characteristics presented in the original source) and provide the reference in which they were mentioned.

Solved by focusing the sentence on chert from the beginning, and not all raw materials. “(…) high potential for chert raw material studies.”

For most of the manuscript we separate between the west and east zone because the cherts are associated with the west and east sub-basin of the Algarve. As mentioned, these basins did not always have the same conditions—it seems important for the authors to respect this division as it provided clear differences in the formations and cherts. In the discussion (especially when discussing the territory used by hunter-gatherers) we no longer apply the western-eastern division since we are no longer referring to geology, but rather human behaviour. We altered the text to make it clearer.

"Line 602. Figure 9, you must put a white background." Solved.

The four formations are: 1) from the west sub-basin, the Lower Jurassic formation (Carixian) and the Upper Jurassic formation (Kimmeridgian); 2) from the east sub-basin, the Middle Jurassic formation (Malhão) and the Upper Jurassic formation (Jordana). The abstract has been corrected.

Line 798. In Flügel's reference, the name of the book is missing (Microfacies of Carbonate Rocks). Solved.

Line 808. In Gómez & Lunt reference, the name of the book is missing (Phylogeography of Southern European Refugia) Solved.

Line 884. Incomplete reference. Solved.

Line 928. Incorrect reference, change Soler B.G by Gómez de Soler, B. Solved.

We appreciate the opinion. It is true that for certain cases, an approximate coordinate would suffice (in the case of secondary deposition settings, for example). However, we believe that data should always be clear and easy to trace. A detailed collection of coordinates, associated with photos of the outcrops and landmarks (and of course, all associated data) provides clear access to the outcrops for future needs or studies, especially when landscapes keep changing.

Reviewer #3:

The authors would like to thank reviewer #3 for the comments, questions and corrections to the manuscript. We believe these were useful and had a positive impact in the scientific quality of the paper.

The authors agree with this commentary, as it is important to refer not only to lithotheques and RM studies outside of western Europe but also within this territory, as they are important references. The manuscript (and corresponding references) has been altered to reflect some of the impactful lithotheques and RM works from France and Iberia. These references include “Sánchez M, Rey M, Rodríguez N, Casado A, Medina B, Mangado X. The LithicUB project: A virtual lithotheque of siliceous rocks at the University of Barcelona. JLS.”; “Ortega D, Terradas X. The lithotheca of siliceous rocks from Catalonia. In: Bostyn F, Giligny F, editors. Lithic Raw Material Resources and Procurement in Pre- and Protohistoric Times: Proceedings of the 5th International Conference of the UISPP Commission on Flint Mining in Pre- and Protohistoric Times”; “Fernandes P, Raynal J-P, Tallet P, Tuffery C, Piboule M, Séronie-Vivien M, et al. A map and a database for flint-bearing formations in Southern France: A tool for Petroarchaeology.”; “Delagnes A, Féblot-Augustins J, Meignen L, Park S-J. L’exploitation des silex au Paléolithique moyen dans le Bassin de la Charente: qu’est-ce qui circule, comment... et pourquoi?”; “Fernandes P, Morala A, Schmidt P, Séronie-Vivien M-R, Turq A. Le silex du Bergeracois: état de la question.”; “Turq A, Faivre J-P, Gravina B, Bourguignon L. Building models of Neanderthal territories from raw material transports in the Aquitaine Basin (southwestern France).”; “Rodríguez AM, Rodríguez JAL, Pelegrin J. The Prehistoric Flint Exploitations of the Milanos Formation (Granada, Spain).”

As reviewer #3 mentioned, Lusolit already had geological samples recovered from the Algarve. However, due to the change of researcher in charge of the Lusolit lithotheque and change in spaces, it was necessary to collect new samples and data regarding the outcrops. This included characterizing the outcrops (through a database and photography) and the recovered chert samples. To obtain consistency in the data, these revisited outcrops and samples were used for petrographic studies, which added a new layer to the previous works of Lusolit, as well as the creation of the online database which currently only includes the samples from 2021/2022, but will be incremented with the previous samples mentioned by Pereira et al (2016b).

The authors expect to make geochemical analyses in the future, for a better elemental characterization of flints and their respective formation environments, as well as to have new comparison features between the geological and archaeological samples (which are often patinated/altered and may not be destroyed). However, as reviewer #3 pointed out, Brandl's (2006) multilayered approach (MLA) specifically mentions geochemical methods. Since these methods are outside the scope of this paper, the authors have removed the reference to the MLA.

line 32: Southwestern instead of Southernwestern Solved

line 33: Cascalheira et al. 2017b appears first here, Cascalheira et al. 2017a only in line 276, change 2017b and 2017a along the manuscript Solved by adjusting the citation methods to [1] in the text and the bibliography by order according to PLOS ONE’s guidelines.

line 53-57: add a reference for Vale Comprido point and its association with the Heinrich Event 2 episode. Solved. Added “Cascalheira J, Bicho N. Hunter–gatherer ecodynamics and the impact of the Heinrich event 2 in Central and Southern Portugal.” 

line 180-201: Triassic and Jurassic formations belong to Mesozoic and not to Paleozoic Solved

line 182: western and eastern (not oriental) sub-basins Solved

---

## [Decision Letter · Decision Letter 1]

1 Aug 2023

PONE-D-22-35505R1Creating frames of reference for chert exploitation during the Late Pleistocene in Southwesternmost Iberia.PLOS ONE

Dear Dr. Belmiro,

Thank you for submitting your manuscript to PLOS ONE. After careful consideration, we feel that it has merit but does not fully meet PLOS ONE’s publication criteria as it currently stands. Therefore, we invite you to submit a revised version of the manuscript that addresses the points raised during the review process.

Please submit your revised manuscript by Sep 15 2023 11:59PM. If you will need more time than this to complete your revisions, please reply to this message or contact the journal office at plosone@plos.org. Please include the following items when submitting your revised manuscript:A rebuttal letter that responds to each point raised by the academic editor and reviewer(s). You should upload this letter as a separate file labeled 'Response to Reviewers'.A marked-up copy of your manuscript that highlights changes made to the original version. You should upload this as a separate file labeled 'Revised Manuscript with Track Changes'.An unmarked version of your revised paper without tracked changes. You should upload this as a separate file labeled 'Manuscript'.If applicable, we recommend that you deposit your laboratory protocols in protocols.io to enhance the reproducibility of your results. Protocols.io assigns your protocol its own identifier (DOI) so that it can be cited independently in the future. For instructions see: https://journals.plos.org/plosone/s/submission-guidelines#loc-laboratory-protocols. Additionally, PLOS ONE offers an option for publishing peer-reviewed Lab Protocol articles, which describe protocols hosted on protocols.io. Read more information on sharing protocols at https://plos.org/protocols?utm_medium=editorial-email&utm_source=authorletters&utm_campaign=protocols.

We look forward to receiving your revised manuscript.

Kind regards,

Enza Elena Spinapolice, Ph.D

Academic Editor

PLOS ONE

Journal Requirements:

Reviewers' comments:

Reviewer's Responses to Questions

**Comments to the Author**

1. If the authors have adequately addressed your comments raised in a previous round of review and you feel that this manuscript is now acceptable for publication, you may indicate that here to bypass the “Comments to the Author” section, enter your conflict of interest statement in the “Confidential to Editor” section, and submit your "Accept" recommendation.

Reviewer #2: All comments have been addressed

Reviewer #3: All comments have been addressed

2. Is the manuscript technically sound, and do the data support the conclusions?

Reviewer #2: Yes

Reviewer #3: Yes

3. Has the statistical analysis been performed appropriately and rigorously? 

Reviewer #2: N/A

Reviewer #3: N/A

4. Have the authors made all data underlying the findings in their manuscript fully available?

Reviewer #2: Yes

Reviewer #3: Yes

5. Is the manuscript presented in an intelligible fashion and written in standard English?

Reviewer #2: Yes

Reviewer #3: Yes

6. Review Comments to the Author

Reviewer #2: This is my second review of the article led by Joana Belmiro et al. entitled “Creating frames of reference for chert exploitation during the Late Pleistocene in Southwesternmost Iberia”. After this revision, I can say that the article is definitely improved and several big issues I had with the previous version have been rectified, which I think definitely benefit the paper. However, as the article is intended to be a framework for future studies of raw materials applied to various archaeological sites in Portugal, some more precise data has to be included. I refer mainly to the geological formations with chert and their reflection in the manuscript, especially in figure 3. For example, in the case of this figure, it has been modified and improved, but since throughout the manuscript the geological formations with chert are mentioned, I think that a more detailed map including the geological formations would be necessary, and not just the series/epoch. I believe this is one of the keys of the manuscript to be a solid framework of reference on chert raw materials for further studies applied to archaeological sites. Nevertheless, the study is a definite improvement.

Regarding the general aspects I now agree with the comments of the authors to continue with the geographic division of western and eastern areas, but I advise to reinforce the idea in geological settings. Just add a sentence mentioning the use of this geographical division because of, as the authors rightly say, the differences in cherts and their formations.

Still on the same topic "frame of reference", although figure 2 has been improved and enlarged, as there are four chert formations, at least one photograph of each formation should be included. Some pictures of chert from Kimmeridgian formations?

Minor issues:

Line 334. When table S3 is cited, it should not be S2, in order to follow an order.

Line 366. At this point, here table S2 should be S3.

Line 378. Maybe it's me that's not working but I can't find the DOI in question.

Line 385. Results. Personally, I think there is still a bit of confusion between geological formation with chert, type outcrop, outcrop and sample. Although figure 3 and table 2 has clarified the data much more, looking at the detailed scale of the outcrops it sounds a bit strange to me that an outcrop, not a formation, has cherts covering a distance of almost 10 km. Wouldn't it be better to have the formation well defined and to have as many outcrops as points where cherts are found in the formation and that each point would take the nearby toponym to define the outcrop/location? This is in line with the general comment at the beginning that the delineation of chert formations greatly strengthens this work as a frame of reference.

It would also be helpful to have the outcrops mentioned in the text with their number and acronym before the figure. This would make it easier to follow the manuscript for those of us who are not familiar with the region.

Reviewer #3: (No Response)

7. PLOS authors have the option to publish the peer review history of their article (what does this mean?). If published, this will include your full peer review and any attached files.

Reviewer #2: No

Reviewer #3: No

---

## [Author Response · Author response to Decision Letter 1]

22 Aug 2023

Reviewer #2:

Reviewer #2: This is my second review of the article led by Joana Belmiro et al. entitled “Creating frames of reference for chert exploitation during the Late Pleistocene in Southwesternmost Iberia”. After this revision, I can say that the article is definitely improved and several big issues I had with the previous version have been rectified, which I think definitely benefit the paper. However, as the article is intended to be a framework for future studies of raw materials applied to various archaeological sites in Portugal, some more precise data has to be included. I refer mainly to the geological formations with chert and their reflection in the manuscript, especially in figure 3. For example, in the case of this figure, it has been modified and improved, but since throughout the manuscript the geological formations with chert are mentioned, I think that a more detailed map including the geological formations would be necessary, and not just the series/epoch. I believe this is one of the keys of the manuscript to be a solid framework of reference on chert raw materials for further studies applied to archaeological sites. Nevertheless, the study is a definite improvement. 

We thank reviewer #2 for their suggestions and comments since they provide valuable feedback and have helped improve this revised version of the article. We believe a map showing the geological formations referenced in the paper is an improvement. We created a new map, showing the chert-bearing formations (not just the series/epoch), with the numbered outcrops where chert can be found within the formation.

Regarding the general aspects I now agree with the comments of the authors to continue with the geographic division of western and eastern areas, but I advise to reinforce the idea in geological settings. Just add a sentence mentioning the use of this geographical division because of, as the authors rightly say, the differences in cherts and their formations. 

A sentence was added to reinforce the geographic division of western and eastern areas in the geological setting.

Still on the same topic "frame of reference", although figure 2 has been improved and enlarged, as there are four chert formations, at least one photograph of each formation should be included. Some pictures of chert from Kimmeridgian formations? 

Two photos of the Kimmeridgian formation outcrop and cherts have been added to the set, to represent all the chert-bearing formations in the Algarve.

Minor issues:

Line 334. When table S3 is cited, it should not be S2, in order to follow an order. We have corrected the SOM files names, to better reflect their reference order. In this case, the Macroscopic analysis dataset is now S2, and the Data dictionaries are now S3.

Line 366. At this point, here table S2 should be S3. Same as previous commentary.

Line 378. Maybe it's me that's not working but I can't find the DOI in question. The short form of the DOI was replaced by the full link “https://doi.org/10.17605/OSF.IO/FP7TA”, which takes the reader to the correct OSF page.

Line 385. Results. Personally, I think there is still a bit of confusion between geological formation with chert, type outcrop, outcrop and sample. Although figure 3 and table 2 has clarified the data much more, looking at the detailed scale of the outcrops it sounds a bit strange to me that an outcrop, not a formation, has cherts covering a distance of almost 10 km. Wouldn't it be better to have the formation well defined and to have as many outcrops as points where cherts are found in the formation and that each point would take the nearby toponym to define the outcrop/location? This is in line with the general comment at the beginning that the delineation of chert formations greatly strengthens this work as a frame of reference. We have clarified the distinction between geological formations with cherts and outcrops throughout the manuscript. The outcrops take the nearby toponym (in some cases, the toponym is the name of the formation, due to it being the name of the area, for example JOR and MALH toponyms), but we have clarified the outcrops to be points where chert is found within the formation.

It would also be helpful to have the outcrops mentioned in the text with their number and acronym before the figure. This would make it easier to follow the manuscript for those of us who are not familiar with the region. Added missing acronyms to the outcrops mentioned in the text, especially before Figure 3 to aid in the comprehension of the location of the outcrops.

---

## [Decision Letter · Decision Letter 2]

10 Oct 2023

Creating frames of reference for chert exploitation during the Late Pleistocene in Southwesternmost Iberia.

PONE-D-22-35505R2

Dear Dr. Belmiro

We’re pleased to inform you that your manuscript has been judged scientifically suitable for publication and will be formally accepted for publication once it meets all outstanding technical requirements.

Kind regards,

Enza Elena Spinapolice, Ph.D

Academic Editor

PLOS ONE

Additional Editor Comments (optional):

Reviewers' comments:

Reviewer's Responses to Questions

**Comments to the Author**

1. If the authors have adequately addressed your comments raised in a previous round of review and you feel that this manuscript is now acceptable for publication, you may indicate that here to bypass the “Comments to the Author” section, enter your conflict of interest statement in the “Confidential to Editor” section, and submit your "Accept" recommendation.

Reviewer #2: All comments have been addressed

2. Is the manuscript technically sound, and do the data support the conclusions?

Reviewer #2: Yes

3. Has the statistical analysis been performed appropriately and rigorously? 

Reviewer #2: Yes

4. Have the authors made all data underlying the findings in their manuscript fully available?

Reviewer #2: Yes

5. Is the manuscript presented in an intelligible fashion and written in standard English?

Reviewer #2: Yes

6. Review Comments to the Author

Reviewer #2: In relation to the revisions made, all the comments have been taken into account. Figure 3 is presented in a clearer way and it is easy to observe the situation of the formations and how they are divided between west and east, an aspect that gives even more sense to the division in the results proposed by the authors. The idea has also been reinforced in the geological settings by the authors.

Regarding the inclusion in Figure 2 of some images of the four formations described in the manuscript, this has also been corrected, as well as all minor issues have been fixed.

An unimportant comment, I have found some minor errors in the bibliography. I believe that with a small check will it be corrected. For example, line 1005 “Gómez de Soler BG” must be replaced by “Gómez de Soler B”, or line 1026 “Arsuaga L”, must be replaced by “Arsuaga JL, or some authors present their compound names separated by dashes and the rest do not... (example line 963 “J-P”, or line 959 “S-J” or line 966 “M-R”)

In short, I believe that changes have improved the manuscript, making it much clearer. After all the work done on the two previous versions I think the manuscript deserves to be published. Congratulations to the authors!

7. PLOS authors have the option to publish the peer review history of their article (what does this mean?). If published, this will include your full peer review and any attached files.

Reviewer #2: No

---

## [Editor Report · Acceptance letter]

12 Oct 2023

PONE-D-22-35505R2 

Creating frames of reference for chert exploitation during the Late Pleistocene in Southwesternmost Iberia. 

Dear Dr. Belmiro:

I'm pleased to inform you that your manuscript has been deemed suitable for publication in PLOS ONE. Congratulations! Your manuscript is now with our production department. 

Kind regards, 

on behalf of

Dr. Enza Elena Spinapolice 

Academic Editor

PLOS ONE